# Effect of a Summer Flood on Benthic Macroinvertebrates in a Medium-Sized, Temperate, Lowland River

**Somsubhra Chattopadhyay** [1], **Paweł Oglęcki** [2], **Agata Keller** [1], **Ignacy Kardel** [1], **Dorota Mirosław-Świątek** [1] and **Mikołaj Piniewski** [1,*]

1   Department of Hydrology, Meteorology and Water Management, Warsaw University of Life Sciences, 02-787 Warsaw, Poland; somsubhra_chattopadhyay@sggw.edu.pl (S.C.); agata_keller@sggw.edu.pl (A.K.); ignacy_kardel@sggw.edu.pl (I.K.); dorota_miroslaw_swiatek@sggw.edu.pl (D.M.-Ś.)
2   Department of Environmental Improvement, Warsaw University of Life Sciences, 02-787 Warsaw, Poland; pawel_oglecki@sggw.edu.pl
*   Correspondence: mikolaj_piniewski@sggw.edu.pl

**Abstract:** Floods are naturally occurring extreme hydrological events that affect stream habitats and biota at multiple extents. Benthic macroinvertebrates (BM) are widely used to assess ecological status in rivers, but their resistance and resilience to floods in medium-sized, temperate, lowland rivers in Europe have not been sufficiently studied. In this study, we quantified the effect of a moderate (5-year return period) yet long-lasting and unpredictable flood that occurred in summer 2020 on the BM community of the Jeziorka River in central Poland. To better understand the mechanisms by which the studied flood affected the BM community, we also evaluated the dynamics of hydrological, hydraulic, channel morphology, and water quality conditions across the studied 1300 m long reach. Continuous water level monitoring, stream depth surveying, and discharge measurements. As well, in-situ and lab-based water quality measurements were carried out between March and August 2020. BM communities were sampled three times at eight sites along the reach, once before and twice after the flood. High flow velocities during the flood resulted in stream bed instability leading to sand substrate movement that caused streambed aggradation by up to 0.2 m. Dissolved oxygen and ammonium-nitrogen were major drivers of BM community structure. Taxa richness, abundance, and the BMWP-PL index declined significantly, whereas Shannon evenness and Simpson diversity indices showed no significant change in the first post-flood sampling, as indicated by Kruskal–Wallis and Tukey tests. Non-metric multidimensional scaling (NMDS) analysis showed that community composition was also significantly affected by the flood. Seven weeks after the flood peak (August 2020 sampling), BM communities had fully recovered from the disturbance. The results can serve as a first approximation of the resistance and resilience of BM communities for relevant applications in other medium-sized, low-gradient, temperate rivers.

**Keywords:** moderate flood; flow-ecology relationships; macroinvertebrate communities; biological diversity; ecosystem resilience; channel bed

## 1. Introduction

Alterations in climate patterns resulting from both natural and anthropogenic processes have raised many significant concerns, including more frequent rainfall events with higher intensity, increasing drought events with higher severity, and rise in seawater levels, which all have further implications on the hydrologic cycle [1]. It is agreed upon in the scientific community that increasing greenhouse gas emissions have been the primary cause of climate change all over the world [2]. Specifically, an increasing trend in temperature across Europe has been observed, while precipitation trends have demonstrated highly variable regional changes without any clear continental impact [2]. More alarming forecasts for several regions in Central Europe exist, as extreme rainfall intensities, particularly for short time periods, have shown an increasing trend over the recent decades [3,4]. As

reported by [3], sub-daily extreme precipitation events will be more intense, accompanied by increasing frequency across Europe in the future. Average annual temperature and precipitation are both anticipated to increase for Poland as reported by [5], with temperature increases between 1 and 4 °C accompanied by an increase of 6–16% in precipitation. Floods are projected to increase in the future, thereby creating a challenge for flood risk reduction, water management, and climate change adaptation for Poland [6].

Aquatic ecosystems are facing increasing stress due to abrupt temporal changes in hydrologic and thermal regimes primarily caused by more extreme, longer, and more frequent low and high flows [7,8]. Benthic macroinvertebrates (BM) have prominent roles in the mosaic of the river ecosystem's structure and are frequently used as indicators of water quality [9]. BM have many advantages as water quality indicators because they are diverse, ubiquitous, can be easily collected, and are sensitive to a range of environmental and chemical stressors [10]. Through before–after event comparisons, researchers have examined the impact of floods on lotic ecosystems and identified flood events as being capable of altering BM communities [11,12]. Effects of flooding with regards to BM communities often include severe modification of stream habitat [13], which further results in directly or indirectly affecting species abundance, altering assemblage composition [14]. Flooding results in increased scouring, transport, and redistribution of sediment and organic matter, which flushes the benthic community further downstream [15]. Moreover, increased shear stress in the streambed during a flood event is another factor that disturbs the association of benthic communities to their preferred microhabitat [13,16,17]. On a contrasting note, floods are also important natural drivers for the aquatic community on stream systems. In a very recently concluded study on multi-decadal effects of floods on BM community structure in the Murray River, [18] reported increasing abundance in all functional feeding groups. The authors attributed this phenomenon to an influx of organic matter of all sizes, from particulate organic matter to coarse and large woody debris following floods, which provide a food resource and/or habitat. A similar conclusion was drawn in the previous work by [19], in which it was observed that floods caused an initial decline in BM abundance and richness, which was succeeded by a sustained increase persisting for over 25 years before returning to pre-flood levels. Piniewski et al. [7] reported that across Europe, macroinvertebrate density and richness responded more negatively compared to fish after floods and droughts. They also found biota resistance to floods to be lower than the resistance to droughts. In general, traits of BM tend to follow evolutional changes of adapting to faster flow; however, many assemblages are still vulnerable to floods [20]. The resilience of BM is highly variable, with recovery times ranging from weeks to several months, as reported in a lowland river in New Zealand [21]. According to [20], benthic community resilience is often determined by the interaction of various factors, such as flood magnitude, disturbance history, effects of the flood on riparian vegetation, food sources, the size of the remaining population, and antecedent conditions.

Due to the unpredictable nature of flood events, research documenting their effects on BM communities has been rare in Poland to date, with one notable exception of a study by [11] of a mountainous stream in the Carpathians. Focusing on temperate lowland rivers, a few studies quantified BM responses to different aspects of flow variability [8,22,23]. None of them, however, dealt with a natural flood event. Grzybkowska et al. [23] studied the effect of high flows (but not floods) on the chironomid community in a medium-sized lowland river in central Poland, reporting a sharp decline in that community following the highest discharge. Szczerkowska-Majchrzak and Grzybkowska [8] investigated the impact of hydrological disturbances of different magnitude on the BM community and found that the highest flow events caused instability of all habitats. The authors found that Oligochaeta were dislodged and dragged downstream before being stopped by macrophytes. A similar observation was also noticed for Chironomidae, the second most abundant BM family in the Drzewiczka River. Dumnicka and Koszalka [22] found an increasing density of Oligochaeta following a hydrological drought in small woodland streams in northeastern Poland.

Considering the complex nature of BM response to floods, which is dependent on a multitude of factors, such as specific watershed and channel attributes, regional studies shed light on the predictive understanding of the relationship between BM assemblages and floods. The present study aims to quantify the effect of a moderate flood event that occurred in June and July of 2020 on the BM community structure in the Jeziorka, a typical, medium-sized lowland river in Poland. In order to obtain a better understanding of the possible mechanisms by which the studied event affected the BM community, we also evaluated the dynamics of hydrological, channel morphology, hydraulic, and water quality conditions in the studied reach, which was often neglected in the previous studies. To achieve the main objective, the following sub-tasks were identified in this study: (a) determining the return period of the flood event that occurred in June and July of 2020, (b) analyzing riverbed stability conditions impacted by the flood event, (c) quantifying the impacts on water quality parameters, and (d) evaluating the flood impacts on BM community structure using abundance, richness, and diversity indices.

## 2. Materials and Methods

### 2.1. Study Area

The Jeziorka River is a tributary of the Vistula, the largest Polish watercourse. The length of the Jeziorka River is 70 km, whereas its catchment area equals 989 km$^2$.

Located in central Poland (Figure 1C), the Jeziorka serves as an example of a typical medium-sized lowland river flowing through a predominantly agricultural landscape. Its course is a mixture of regulated and semi-natural reaches, with the visible influence of human activity within the catchment, such as dykes, fish ponds, and groundwater abstraction sites. The average annual rainfall is 632 mm, and the average annual streamflow at the nearest IMGW-PIB (the Institute of Meteorology and Water Management–National Research Institute) flow gauge Piaseczno 2 (Figure 1B) is 2.7 m$^3$/s. As in the case of most temperate lowland rivers in Central Europe, summer months (June, July, August) are usually the wettest. The seasonal streamflow distribution features a peak in early spring due to low evapotranspiration rates in winter and melting snow cover in early spring. Diverse aspects of human activity in the catchment and habitats' structure (meadows, forests, and oxbow lakes) also impact the biological diversity of the valley bottom [24]. Water resource management in the Jeziorka catchment is facing challenges due to high sub-urbanization pressure that is expected to grow in the future in the outskirts of Warsaw, as well as pressure on water resources from irrigation, fish farms, and municipal users [25]. Table 1 displays information about various abiotic features of the catchment.

**Table 1.** Selected abiotic characteristics of the Jeziorka River.

| Variable | Mean | Range |
|---|---|---|
| Channel width [1] (m) | 7 | 5–10 |
| Channel depth [2] (m) | 0.6 | 0.3–1.15 |
| Dominant substrate [1] | Mud, sand | |
| Catchment area (km$^2$) | Whole catchment: 989; Flow gauge Piaseczno: 683; Studied reach: 382 | |
| Mean annual flow [3] (m$^3$/s) | 2.7 | 0.1–58.3 |
| 100-year flood [3] (m$^3$/s) | 51.9 | |
| 10-year flood [3] (m$^3$/s) | 32 | |

[1]—for the reach studied in this paper; [2]—cross-sectional mean depth from a number of measurements taken at site M8; [3]—for the IMGW-PIB flow gauge at Piaseczno (cf. Figure 1).

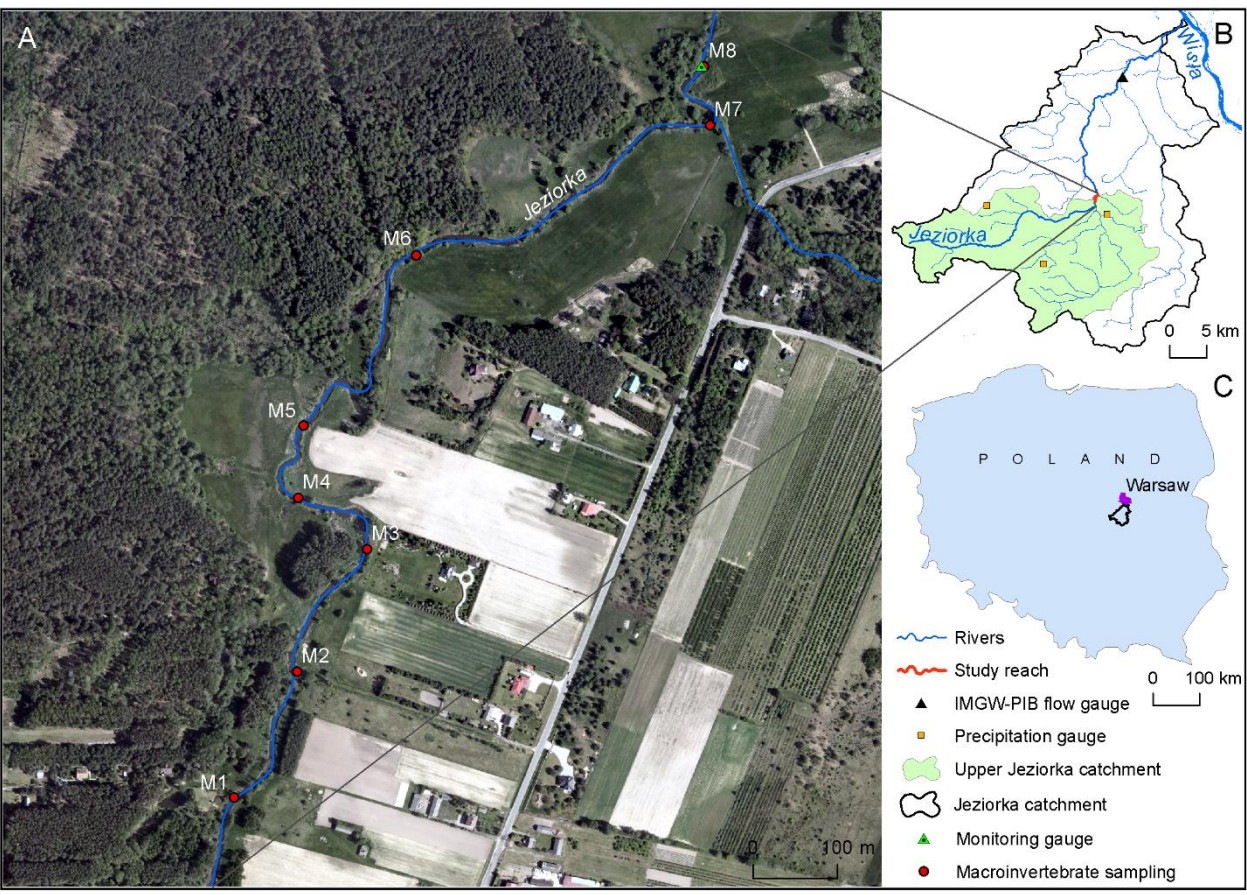

**Figure 1.** Study area: (**A**) selected reach of the Jeziorka River with eight benthic macroinvertebrate (BM) sampling sites and a flow and water quality monitoring gauge against an orthophoto map; (**B**) location of the reach within the Jeziorka catchment; (**C**) location of the Jeziorka catchment in Poland.

### 2.2. Flood Event

The flood event that occurred in the Jeziorka in summer 2020 began with rainfall that started on 17 June and gained in strength over time. Rainfall occurred every day until 24 June. On 22 June, a heavy storm caused a flash flood in urban areas of the catchment (https://piasecznonews.pl/tona-ulice-posesje-zmieniaja-sie-w-jeziora/, accessed on: 14 October 2020), and the river flooded the floodplains in the following days. The water level at Piaseczno 2 gauge exceeded the warning stage and approached the alarm stage on 28 June with a 4-cm margin. By 10 July 2020, water levels reached the pre-flood conditions. The rest of July and the entire month of August were warm and dry, which resulted in low water levels during these months. Figure 2 depicts the river channel at the most downstream measurement site M8 (Figure 1A) during the flood peak on 25 June and 7 weeks later, on 12 August, during the low-flow conditions. The event was also documented using a short video recorded at site M8 on 25 June 2020 (Supplementary Material, Video S1).

For the purpose of this research, a study reach of 1300 m in the middle course of the Jeziorka was selected (Figure 1A). The upstream catchment area of the downstream end of this reach equals 382 km². Stream channel width at bankfull flow ranges from approximately 5 to 10 m (Table 1). It is a semi-natural reach, with meandering sections, no dykes, and a natural floodplain. Its hydromorphological state was assessed as good using the river habitat survey method [26].

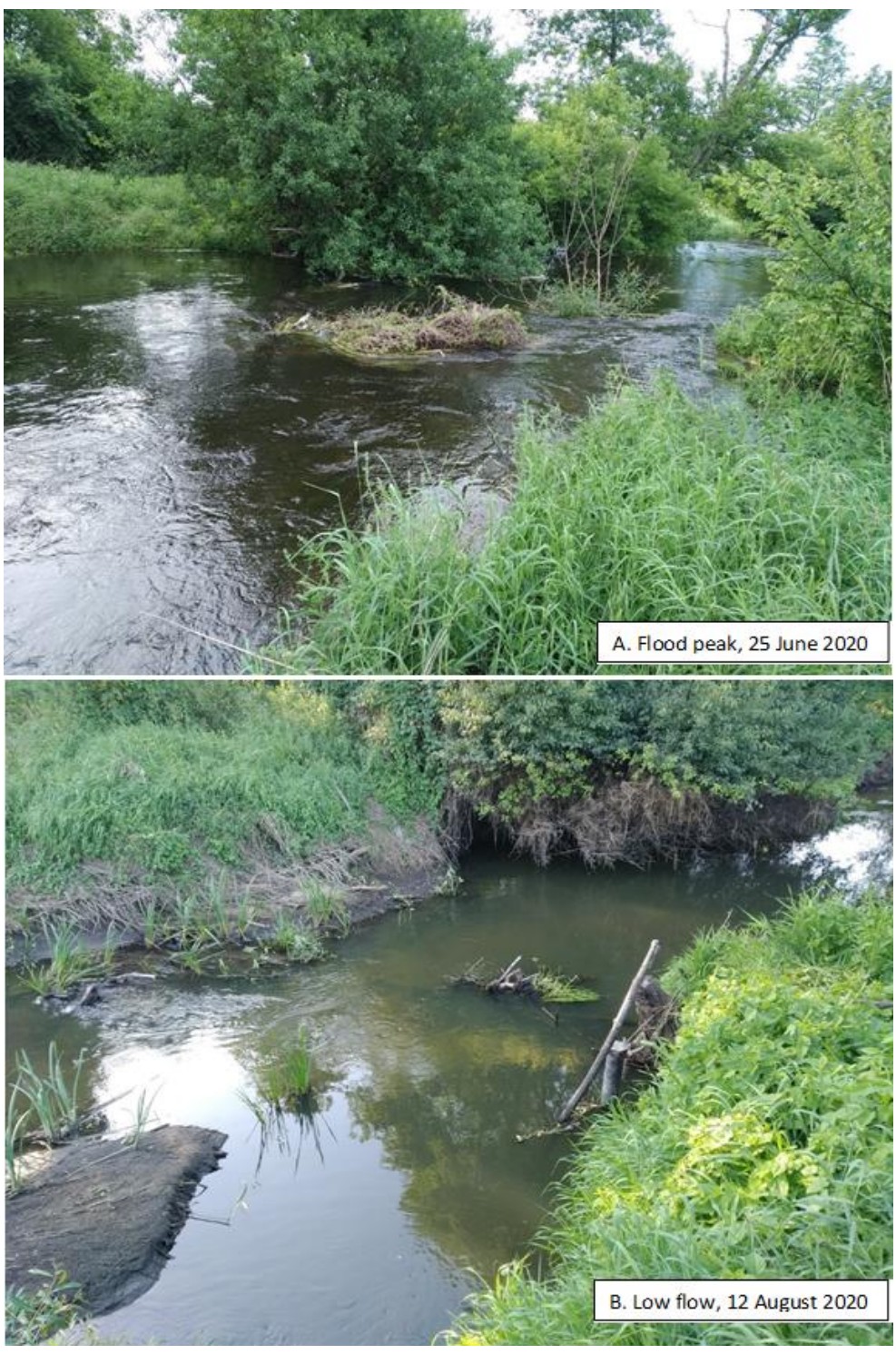

**Figure 2.** Pictures showing the measurement site M8 at the Jeziorka River (**A**) during the flood peak (25 June 2020) and (**B**) during the low-flow period (12 August 2020). The difference in measured water levels between these dates was 1.8 m.

*2.3. Environmental Monitoring*

2.3.1. Hydrological Monitoring

In order to continuously monitor hydrological conditions in the study reach, we set up a flow gauge at the downstream end of the reach (site M8 in Figure 1A). A Solinst water level datalogger was installed in a piezometer located in the stream bed and started recording water levels with an hourly time step on 26 March 2020. Discharge measurements were

performed 12 times between 26 March and 8 October 2020, using OTT MF Pro flowmeter with an absolute pressure depth sensor (accuracy being a larger of two values: $\pm 2\%$ of measured value or $\pm 0.015$ m) and an electromagnetic velocity sensor (accuracy being a larger of two values: $\pm 2\%$ of measured value $\pm 0.015$ m/s), enabling automatic calculation of discharge using the EN ISO 748 method. The measurements were carried out for flows ranging from 0.43 to 6.43 $m^3$/s, which facilitated the development of a stage-discharge rating curve using the power function. Velocity measurement results also allowed for developing stream velocity contour profiles for different flow conditions and for estimation of critical velocities for bedload transport. Surfer software was used to draw the velocity contour profiles. Measured cross-sectional profiles were used for evaluating changes in channel bed structure caused by the flood. Hourly precipitation data were available from three automatic weather stations located in different parts of the catchment area (Figure 1B), and an aerial average was used to correlate with the discharge and water level measurements.

### 2.3.2. Water Quality Monitoring

Physicochemical parameters of water (temperature, pH, electrical conductivity for 25 °C (EC), and dissolved oxygen (DO)) were determined directly in the field using the YSI Professional Plus probe. The water quality analysis was conducted in the Surface Water Monitoring Laboratory, Water Centre of the Warsaw University of Life Sciences. Integrated grab water samples were collected using a sampling bucket from different depths without triggering bottom sediment and stored in two Polypropylene (PP) containers (60 mL and 1500 mL). The containers were transported to the laboratory in a car fridge and analyzed on the same day. The 5-day biochemical oxygen demand ($BOD_5$) was determined using a volumetric method on the OxiTop Control WTW. The orthophosphates ($PO_4$) were detected on a UV-VIS Helios Alpha Spectrometer using ammonium molybdate reagent (ISO 6878). The other dissolved ions in water were determined with a liquid chromatography method (PN-EN ISO 14911 for cations and PN-EN ISO 10304-1 for anions), and the equipment manufactured by Dionex ICS-1000 was used (only before these analyses samples were filtered through 0.45 $\mu$m filters). Total suspended solids (TSS) were determined using the weight method in accordance with PN-EN 872:2007.

Grab samples were collected 11 times during the monitoring period (26 March–19 August 2020) at site M8. Basic physicochemical parameters were also measured twice in other monitoring points, showing negligible differences from M8, which justifies the use of only one monitoring site. Samples were taken between one and three times per month at different flow conditions. Seven samples were taken in pre-flood conditions, two during, and two after the flood event.

### 2.3.3. Monitoring of Benthic Macroinvertebrates

BM samples were collected three times during the study period (on 29 March, 19 July, 21 August) from eight sites selected along the study reach (Figure 1A) in the upper Jeziorka. Sampling sites were selected to capture habitat heterogeneity within the whole reach. A reference sampling characterizing pre-flood conditions was made on 29 March, whereas subsequent samplings were performed on 19 July and 21 August, three and seven weeks after the flood peak, respectively. These datasets allowed us to investigate the effect of flooding on BM communities, and in particular, their post-flood recovery. Sampling focused on determining the composition of the assemblages and the densities of particular taxa. In each cross-section, samples were collected at the site representing dominant, visually identified physical habitat conditions (such as water depth, bottom substrate, and flow velocity). At each site, BM were collected from 1 $m^2$ of the riverbed. Sampling was conducted with 60 $\mu$m triangular dip net, Ekman grab, mosquito dipper, and tweezers, depending on habitat conditions in particular sites, in order to collect as many taxa and specimens as possible. The taxa were identified in the laboratory, partly from non-preserved material within 2–3 days after the sampling and partly from the samples preserved with

70% ethanol. All the specimen were identified to the lowest unquestionably identifiable taxonomic level, with the help of specialist guides [27,28]. The sampling protocol was mostly consistent with those reported in [29–31].

Due to a short distance between sampling sites and lack of major tributaries within reach, the environmental conditions were mostly homogeneous. All sampled sites were characterized by the presence of mud, silt, and sand and occasionally submerged or decaying vegetation and gravel. The substrate was analyzed only on a qualitative basis, with the exception of site M8, for which grain size distribution was analyzed for the purpose of channel bed stability estimation.

*2.4. Data Analysis*

2.4.1. Hydrological Variation

The origin of the flood of summer 2020 in the Jeziorka catchment was a long-lasting rainfall with periodically heavy intensities (Figure 3). Peak hourly rate measured at a single station was 27 mm/h on 22 June. However, due to the local occurrence of particular storms, the maximum areal rainfall rate in the catchment did not exceed 12 mm/h. The maximum daily areal rainfall amounted to 46 mm on the same day, whereas the cumulative areal rainfall between 17 June and 3 July was 168 mm.

Three independent peaks can be distinguished during the whole studied event (Figure 3). The first peak with the maximum hourly flow reaching almost 8 m$^3$/s was the highest and lasted the longest (full three days starting on 24 June). The second peak was small, whereas the third peak was most likely associated with the saturation-excess overland flow, in which most of the rainfall contributes to surface runoff.

Flow conditions on the first (pre-flood) and second (post-flood) sampling dates were similar, with flows ranging between 0.7 and 0.8 m$^3$/s. On the third date, the discharge decreased below 0.5 m$^3$/s, which is low for a river of this size but does not imply drought conditions. Both the cross-sectional area and measured stream velocities changed significantly during the flood (Figure 4). While velocities during normal flow conditions did not exceed 0.4 m/s, in early July 2020, they exceeded 0.65 m/s. Velocity distribution in August during low flows was the most significantly affected by macrophytes, which were absent in March.

2.4.2. Bed Stability Estimation

River morphology and bed stability are determined with non-fluvial deposits with a broad range of grain size distribution [32]. The critical velocity criterion was used to analyze bed stability in the Jeziorka River. Bedload grains are stable as long as the mean velocity is less than critical velocity. After exceeding critical velocity, bedload movement is initiated. Van [33] derived the critical depth-averaged velocity from the critical bed-shear using the Chezy equation and formulated the critical depth-averaged velocity $v_{cr}$ for sand particles in the range of 0.0001 to 0.002 m in the following form:

$$v_{cr} = 0.19d_{50}^{0.1}log\left(\frac{12h}{3d_{90}}\right) \quad \text{for } 0.0001 \leq d_{50} \leq 0.0005 \text{ m} \tag{1}$$

$$v_{cr} = 8.5d_{50}^{0.6}log\left(\frac{12h}{3d_{90}}\right) \quad \text{for } 0.0005 \leq d_{50} \leq 0.002 \text{ m} \tag{2}$$

where:

$d_{50}$ = median particle diameter (m)
$d_{90}$ = 90% particle diameter (m)
$h$ = mean water depth (m)

In order to characterize the substrate conditions, bed sediment was collected at site M8. Samples were collected along a straight section of the river, from both sides of the river, and at several locations in the cross-section, and were averaged by mixing. Sediment

samples were dried, weighed, and sieved at standard intervals to determine the grain size distribution, and $d_{50}$ and $d_{90}$ were calculated.

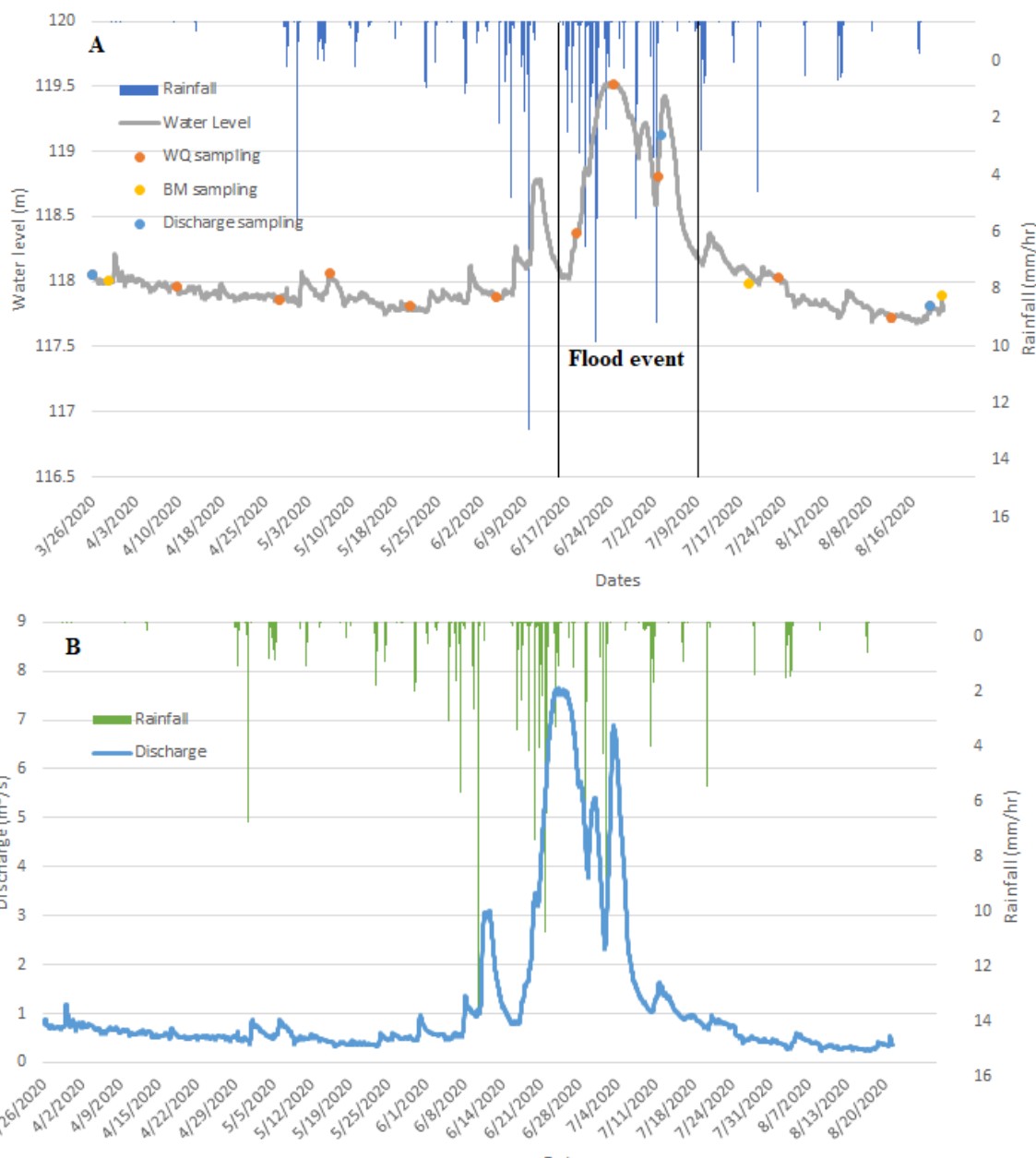

**Figure 3.** (**A**) Comparison of rainfall and water level from March to August 2020 at site M8 highlighting the start and end of the flood event as well as the sampling dates for water quality and macroinvertebrates. (**B**) Time series of daily discharge and rainfall at site M8 from March to August 2020.

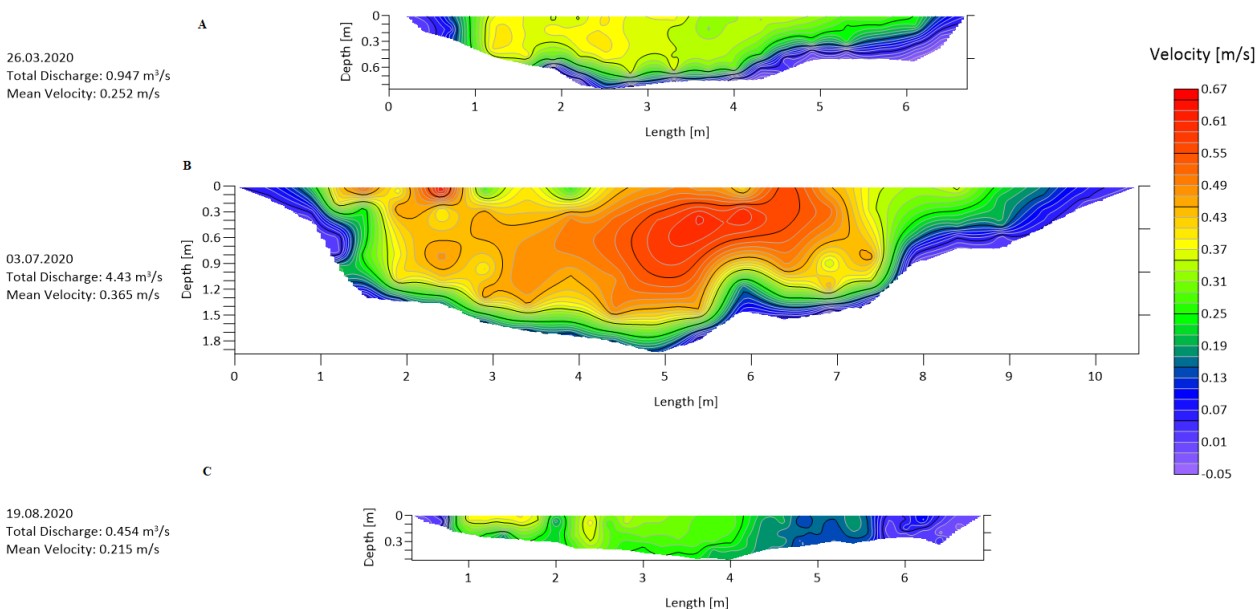

**Figure 4.** Velocity contour profiles during (**A**) normal flow conditions in March, (**B**) post-flood conditions in July, and (**C**) low-flow conditions in August.

### 2.4.3. Analysis of BM Indices

To characterize the diversity of the BM community, three indices, namely taxonomic richness, Shannon evenness index (E), and Simpson diversity index (D), were used in this study. The E index was based on the Shannon–Wiener diversity index (H) in the following way:

$$H = -\sum\left(\frac{n_i}{N}\right)\ln(n_i/N)$$

$$E = H/\ln(s)$$

where $n_i$ is the number of individuals of ith species, N is the total number of individuals, and ln (s) is the natural log of the total number of species recorded. The Simpson diversity index (D) was calculated according to the equation below:

$$D = 1 - \left(\sum n_i(n_i - 1)/N(N - 1)\right)$$

In order to assess water quality from the ecological health perspective, we used the BMWP-PL index, a Polish version of the biological monitoring working party score system (BMWP). Initially designed as an index reflecting river water quality status, it was later reported to also denote physical habitat degradation, and thus, serves as an indicator of ecological status [31]. All sites were assigned a specific ecological quality class based on the BMWP-PL score.

In addition, non-metric multidimensional scaling (NMDS) was performed to determine if the community composition underwent a change because of the flood event. The metaMDS function in the vegan package [34] was used to visualize the community structure using NMDS. Abundance data were cubically transformed, and similarities in taxonomic composition between surveys were analyzed using Bray–Curtis distances.

We analyzed the significance of differences in E, D, and BMWP-PL indices with the non-parametric Kruskal–Wallis test and Tukey test of differences in means using a significance level of $\alpha = 0.05$.

## 3. Results

### 3.1. Flood Frequency Analysis (FFA)

In order to estimate the return period of the flood event analyzed in this study, we used 30 years of daily flow data (1988–2018, with data missing for the year 1997) from the Piaseczno 2 gauge located 24 km downstream from the study reach. We used the peak-over-threshold (POT) method, as several studies have highlighted its advantage over the more conventional annual-maximum-flow method [35,36]. Following evaluation of the statistical parameters of the 30-year time series, it was decided to choose a threshold of 13.5 m³/s and set the parameter characterizing the minimum break between two independent events to 3 days. As a result, a total of 52 independent flood events were selected (Figure 5).

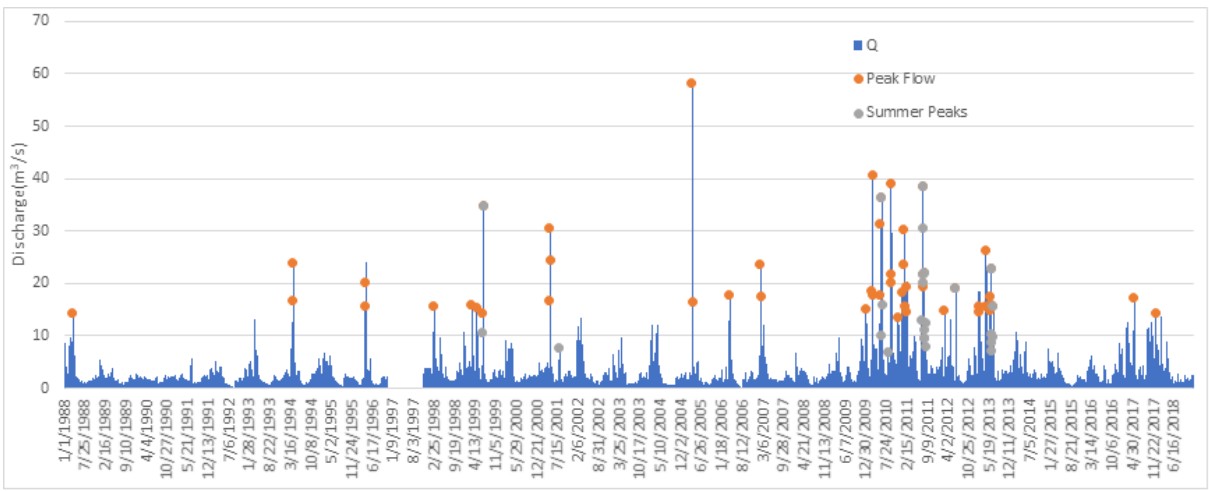

**Figure 5.** Time series of daily discharge at the IMGW-PIB flow gauge Piaseczno 2 from 1988 to 2018, with the selected peaks highlighted for the flood frequency analysis.

Corresponding to the highest flow of 24.9 m³/s that was recorded on 25 June 2020, the return period of this particular flood event was determined to be five years. Since spring floods, generated as a result of snowmelt, are dominant in the POT time series, we carried out an independent FFA restricted to summer flows only for which flood generation mechanism is rainfall. A threshold of 7.04 m³/s was chosen for this purpose (Figure 5). In this case, the return period of the event was close to 10 years, and this was the highest summer peak since 2010.

### 3.2. Assessment of Bed Stability

Analysis of grain size distribution at site M8 indicated $d_{50}$ and $d_{90}$ to be 0.00023 and 0.00045 m, which corresponds to fine and medium sand, respectively. Table 2 summarizes the calculated mean water depth, mean velocity, and critical velocity in the riverbed zone at site M8.

**Table 2.** Calculated mean and critical velocity in riverbed zone (site M8; $v_m$—mean velocity; $v_{cr}$—critical depth-averaged velocity; $h_m$—mean water depth).

| Date | $h_m$ (m) | $v_m$ (m/s) | $v_{cr}$ (m/s) |
|---|---|---|---|
| 26.03.2020 | 0.65 | 0.277 | 0.309 |
| 03.07.2020 | 1.45 | 0.388 | 0.338 |
| 19.08.2020 | 0.37 | 0.249 | 0.289 |

The results show that the critical velocity, which is a function of water depth, varies from 0.289 m/s to 0.338 m/s at depths varying from 0.37 m to 1.45 m. As indicated in Table 2, only under the flood conditions, mean velocity exceeded critical velocity resulting

in substrate movement that can lead to disturbances in the BM community structure. On the other hand, the bed stability conditions ($v_m < v_{cr}$) were fulfilled for the March and August sampling.

The analysis of changes in channel bed morphology over time in site M8 suggests that the flood has caused substantial changes in it (Figure 6). Channel bed elevation for each of five dates was estimated based on cross-sectional water depth readings taken during discharge measurements and stream water levels recorded using the data logger. The differences in channel bed shape were negligible for two pre-flood and two post-flood measurements and can be attributed to measurement errors. However, the channel bed profile measured during flood flows on 3 July suggests the occurrence of scour in the central part in which velocities were the highest. Comparison of channel bed elevation between 3 and 24 July demonstrates that fill occurred. The layer of deposited bed material ranged between 0.2 to 0.4 m.

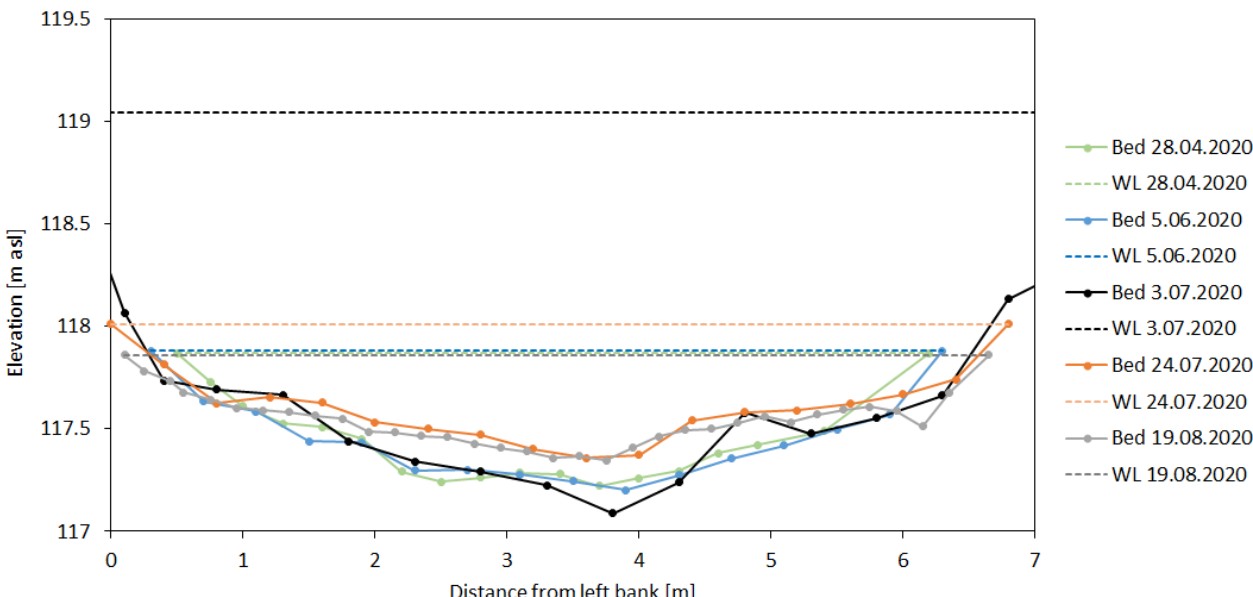

**Figure 6.** Channel bed morphology before (28 April and 5 May) and after (24 July and 19 August) the flood event at site M8. Solid lines show the cross-sectional channel bed elevation, and dashed lines show water level (WL) elevation on a given date.

### 3.3. Changes in Water Quality

In addition to hydrological variation, changes in water quality parameters were also analyzed in site M8 (Figure 7). EC values remained relatively stable during the study period, although two measurements after the flood showed significantly higher values. DO levels, expressed in percent saturation of oxygen, were the highest during the March sampling and steadily declined afterward. The two lowest values (below 30%) occurred during the flood event, whereas after the flood, DO increased to approximately 40%. The pH levels were the lowest during the flood, reaching a value of 7, which was followed by an increase to approximately 8 after the flood. TSS concentrations exhibited some fluctuations during the study period, but the samples taken during the flood event did not differ from those before the flood. However, after the flood, the lowest value (below 20 mg/L) was identified. Just prior to the flood event, high concentrations of $NH_4$-N were recorded. High discharge during the flood led to a sharp decrease of $NH_4$-N concentrations, whereas after the flood, an increasing trend occurred. A contrasting behavior was noted for $NO_3$-N, whose concentrations sharply increased during the flood and decreased afterward. $PO_4$-P concentrations exhibited irregular fluctuations, and there is little evidence for any effect of the flood event. Finally, $BOD_5$ concentrations dropped significantly on the peak flow date

in June, reaching only 2 mg/L. However, values in this range were also noted in April and in August.

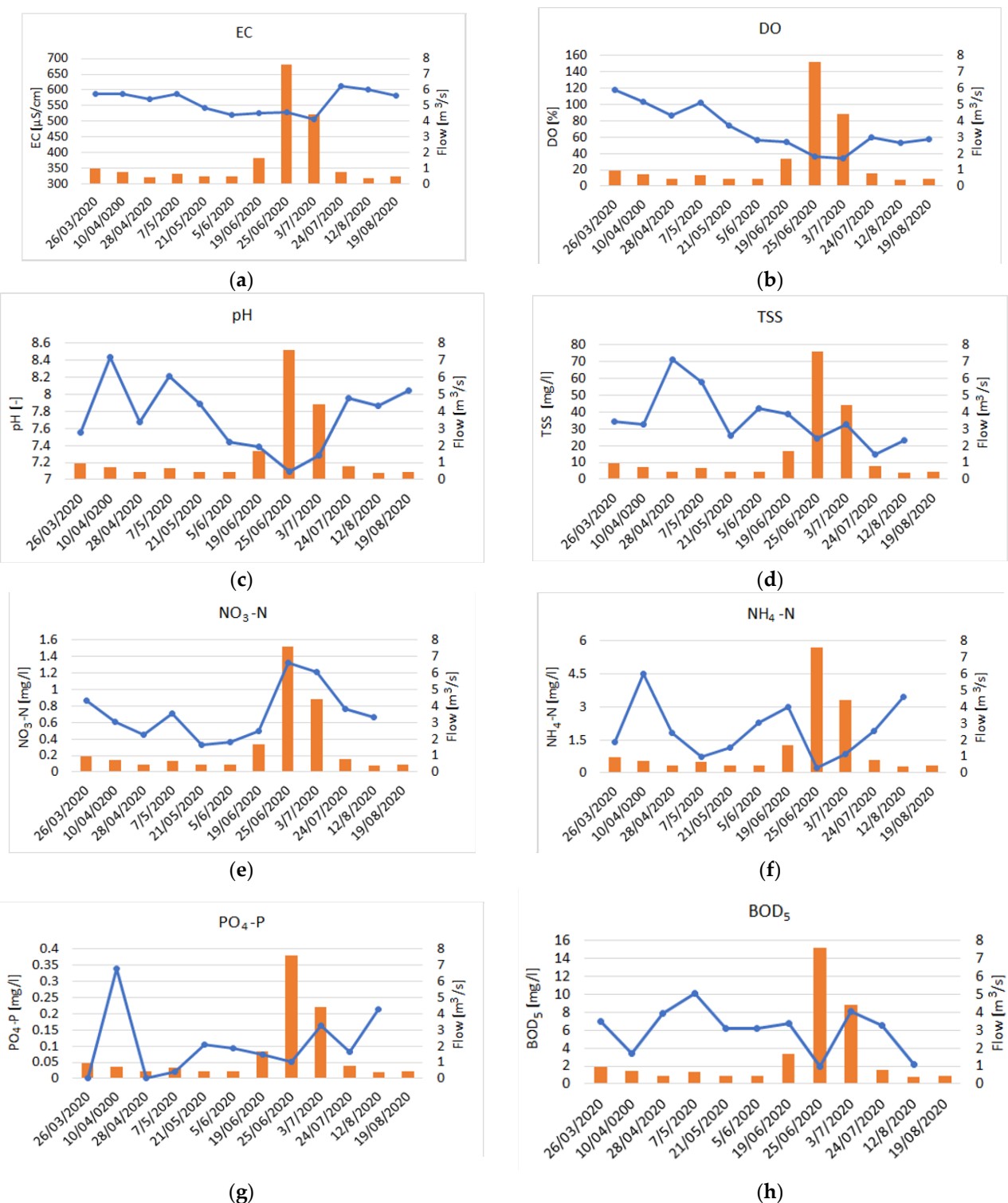

**Figure 7.** Variations in water quality parameters (**a**) EC (**b**) DO (**c**) pH (**d**) TSS (**e**) NO$_3$-N (**f**) NH$_4$-N (**g**) PO$_4$-P (**h**) BOD$_5$ at site M8 from March to August 2020. Orange bars represent measured or calculated flow values.

### 3.4. Changes in BM Community

Abundance data suggest that when averaged across sites, the BM population, which was 66 in the March survey, declined to 38 individuals found in the July survey (Figure 8A and Supplementary Material, Table S1). However, the abundance levels again reached the pre-flood level, with an average of 68 specimens observed during the August 2020 sampling phase. The mean abundance across sites was significantly larger during the March 2020 sampling than during the July 2020 surveys based on the Kruskal–Wallis rank-sum and Tukey comparison tests. A similar trend was observed with mean abundance data of August 2020 surveys significantly higher than July 2020 surveys, although no such difference with March surveys was noticed, according to Kruskal–Wallis and Tukey tests. In general, high abundance was observed at site M2, which had predominantly gravel substrate. Both sites M2 and M7 were impacted heavily; however, a significant recovery was noted at both sites. Woody debris and decaying vegetation were a common feature at these sites, which were providing refuge to BM. High flows caused dislodging, but in summer, macrophytes grew quickly, which may have driven the recovery at these sites.

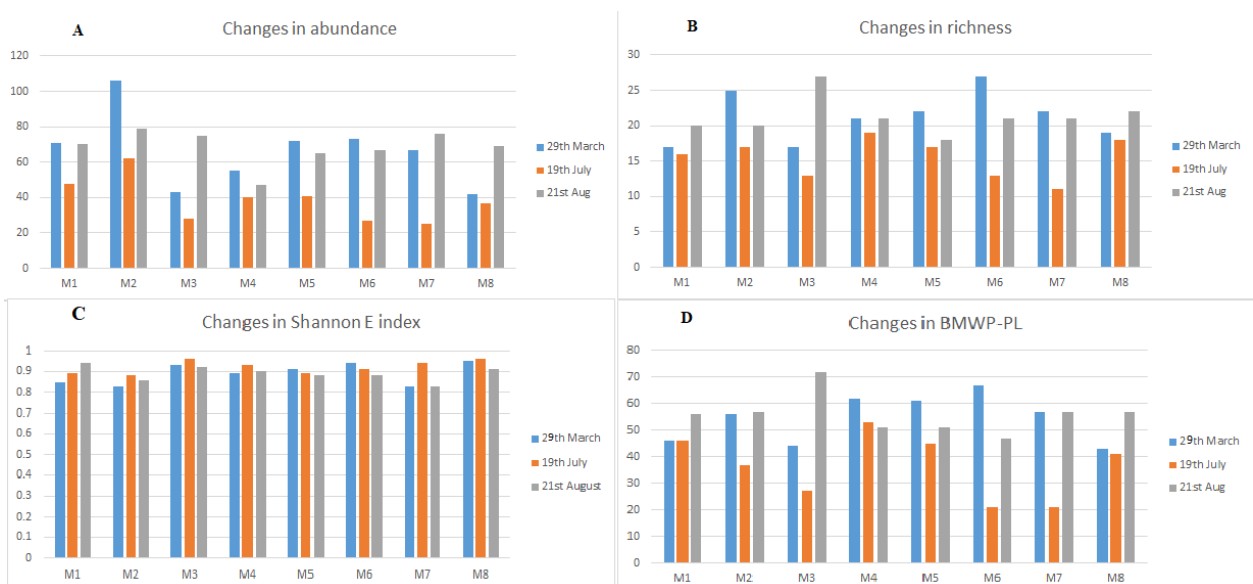

**Figure 8.** Spatial variation of (**A**) abundance, (**B**) richness, (**C**) Shannon evenness index (E), and (**D**) BMWP-PL for the three sampling dates.

The taxonomic richness of the studied reach (calculated as cumulative richness across 8 sites) declined from 45 to 28 after the flood event and again recovered to 41 during August. The highest reduction after the flood occurred in site M7, where the number of taxa decreased from 22 to 11 (Figure 8B). The Kruskal–Wallis rank-sum test results revealed that taxa richness at all sites was significantly lower after the floods (July 2020) than before the floods (March 2020). The Tukey test results were also consistent in the sense that mean richness in March 2020 was significantly higher than the reported value for July 2020. Mean richness was found to be again significantly higher in August 2020 than it was in July 2020, implying that taxa richness returned to the pre-flood values during 4 weeks after the July sampling.

Figure 8C shows that the highest value of the Shannon evenness index (E) was recorded at both sites M3 and M8 (0.96) during the August 2020 survey. The range of variation for both E and Simpson index (D) was similar with a relatively narrower range for the Simpson index (not shown). The lowest value of E was observed at site M2 during the March 2020 sampling. In the pre-flood sampling, the highest value of the Simpson index was noted in site M6 (0.96). There were no statistically significant differences between the mean E and D indices from any survey period, based on the results of the Kruskal–Wallis

and Tukey tests. Nevertheless, both diversity indices seemed to have increased slightly after the floods in the July 2020 survey compared to the March 2020 survey. A possible explanation of this observation is that these indices also count for evenness in the sample, and although the number of taxa was reduced in the July survey, perhaps they were distributed more evenly.

According to the BMWP-PL criteria, all the sites were classified in class III of water quality during the pre-flood sampling (Figure 8D). The average score of BMWP-PL was 54, 36, and 56 in March, July, and August, respectively. Thus, from a moderately impacted stream, the ecological state degraded to a polluted or impacted state in July 2020. The BMWP-PL scores of the flood-impacted stream were found to be significantly different from the pre-flood values and the recovery phase as determined from the Kruskal–Wallis test. Ecological status showed a gradual improvement, as the highest BMWP-PL scores were recorded in August. The BMWP-PL values seemed to be heavily influenced by the flood event, which is in agreement with findings reported earlier for a mountainous stream in Poland [11]. As described above, taxa richness was also found to be responsive to flood events, and as it is the basis for BMWP-PL calculation, both of these scores potentially reflect the flood event impact.

The upper Jeziorka is characterized by a diverse assemblage of BM species that display various life histories and trait-based adaptations, such as swimming, feeding habits, or substrate attachment, which help to increase resistance and resilience to extreme hydrological events. Our results demonstrated that floods impacted almost every species; however, few of them were mainly responsible for the initial recovery of the ecological community. Taxa such as *Tubifex* sp. and *Gammarus* sp. were more abundant in low-flow conditions, which is a preferred habitat for limnophilic taxa (see Supplementary Material 2). These taxa were clearly demonstrating lesser abundance after the flood event, whereas rheophilic species such as *Ephemera vulgata* became more abundant.

Another pattern emerged out for some taxa such as *Dendrocoelum lacteum* and *Cheumatopsyche lepida,* which remained virtually unaffected by the flood and remained present throughout the surveys. Some members of the Coleoptera family, such as *Hydrobius fuscipes* and *Gyrinus* sp., preferred low flows and were drastically reduced from the July samplings. A similar observation was made for Gastropods, which were most abundant during the low-flow conditions but were notably low (almost absent in most of the sites) during the high flow conditions following the flood. Community composition from July and August clearly differed with distinct clustering as evidenced by the non-metric multidimensional scale ordinations presented in Figure 9. Importantly, the taxa composition in the post-flood July sampling was clearly more clustered than both the pre-flood and the second post-flood samplings. Species composition in March was the most dispersed, although it resembled, to some extent, that from August. This pattern differed across sites, e.g., in sites M2 and M4, taxa composition was similar in March and August, whereas in sites M3 and M5, it was quite different. While the substrate type was similar in M2 and M4, M3 and M5 differed slightly in terms of vegetation and presence of macrophytes, which is a possible explanation for this finding.

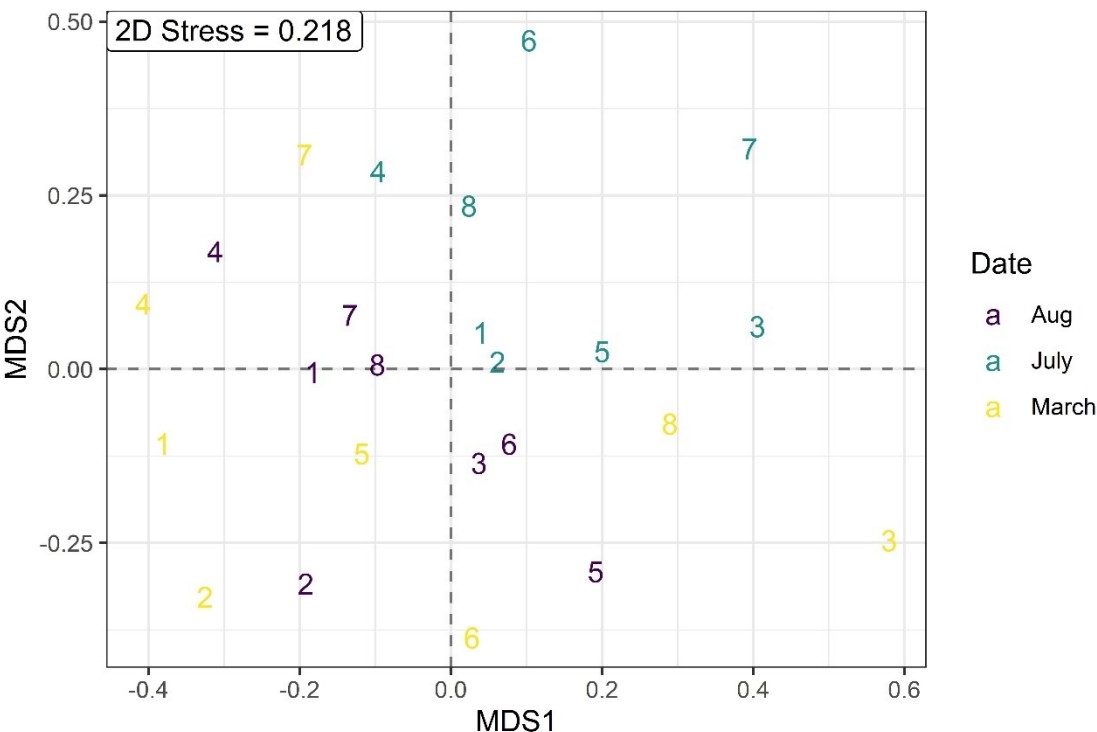

**Figure 9.** Non-metric multidimensional scaling ordination based on a matrix of abundance data for all taxa collected from 8 study sites in the Jeziorka River during three surveys.

## 4. Discussion

BM typically have an optimum flow velocity ranging from 0.3 to 0.7 m/s [37], and for this study, flow velocity in general was found to be suitable for them. BM community composition may have been impacted by velocity fluctuations in this study. Faster flow velocity helps in-stream aeration and organic matter transport, which reduces the effects of river pollution on BM communities [38,39]. In contrast, faster water velocity could be detrimental to BM communities by washing away the food resources and altering the substrate composition [40,41]. For sandy lowland streams in the Northwest European region, [42] noted that a flow velocity of 0.6 m/s, typical of peak discharge conditions, is critical for all species. Our results seem to be confirming this observation as, during the flood event, dislodgment may have occurred, preventing the species from returning to the bottom.

Our results show that conditions for substrate movement and thus loss of bed stability for BM community occurred on 3 July when the measured discharge was equal to 4.43 m$^3$/s. However, the peak discharge was estimated at 7.6 m$^3$/s already on 24 June and lasted about three days, so it can be safely assumed that unstable conditions of the channel bed could have lasted even for two or three weeks (cf. Figure 4C). Our discharge measurements did not capture the peak flow when habitat stress and displacement of invertebrates must have been larger than measured on 3 July. This relatively long duration of the flood might be even more important from the ecological viewpoint than its peak due to a long-lasting disturbance of the stream bed [43]. Shaw et al. [44] showed that for rivers with sandy streambed, major flood events have the potential to aggrade and degrade the streambed to depths of several decimeters, with aggradation taking place during low flows. Findings from this study also indicate that the filling process after the flood resulted in up to 0.2 m higher channel bed. A study by [45] investigated the changes of total invertebrate density and taxonomic richness under conditions of loss of bottom stability determined by exceeding the critical value of the bed-shear stress, described by the Shields entrainment function. The authors showed that total density and taxonomic richness declined at shear stress, exceeding the critical Shields value for substrate mobility. The maximum taxonomic

richness occurred at low velocities and in highly stable sediment. In this study, the concept of critical depth-averaged velocity instead of critical bed-shear stress was used, because the collected data allowed substratum stability assessment using only this parameter.

Our results highlighted that flow velocity played an important role in BM distribution. Naididae and Gammaridae preferred low flows. In contrast, Ephemeridae demonstrated a higher preference for faster velocity and peak flow rates, probably because of the higher oxygenation rates [46].

The severity of the flood event, in general, has reduced the overall richness and abundance in the studied reach. It was apparent that the flood had the most prominent impact on taxa abundance, as it decreased on average by 42% after the flood. Our results are in agreement with earlier studies that have reported significant declines in the abundance of BM after flood events [47–50]. BM abundance can decline by as much as 70–95% succeeding a flood event [7]. In many similar studies on BM responses to floods, sampling was conducted within two weeks of the flood event. It must be kept in mind that our surveying was conducted three weeks after the flood peak, which may have allowed some recolonization from upstream source areas. This may be a probable reason for the relatively small decrease in abundance between the March 2020 and July 2020 sampling. Another source of bias is that we do not know the state of the macroinvertebrate community directly before the flood event but three months before.

As expected, in a similar way, taxa richness was also impacted by floods with an average decrease of 27%, consistent with earlier studies [47,49,51–53]. Some of these studies, however, reported an extreme decline in taxa richness, amounting to a 95% loss in taxa richness after a large flood in northeastern Kansas [54]. Post-flooding community recovery timings could be fast; however, different taxa groups may have different recovery times, and smaller streams are impacted more heavily than larger streams [55]. While recovery time is dependent on various factors, the flood return period is an important one, and it typically equals one month for a flood event with a 20% annual exceedance probability [17]. The flood event in June, which had a return period of 5 years (if all-year time series are analyzed) and almost 10 years (if the analysis is restricted to summer months), showed that recovery of the BM community after the flood of a similar return period lasted one month, which is consistent with our findings.

According to the intermediate disturbance hypothesis (IDH), maximum species diversity occurs when the ecological disturbance is not too rare and not too frequent. Pertaining to flood events and macroinvertebrate community structure, [56] reported that more frequent rainfall events might lead to a long-term decrease of aquatic organisms, but rainfall intensity has a greater influence on the species richness and abundance of the benthic macroinvertebrate communities than that of rainfall frequency. In Jeziorka, this moderate flood event could be categorized as a medium ecological disturbance for which the recovery time was fast, and more importantly, diversity was not impacted, and in some instances, increased following the disturbance. Kim et al. [57] found that in a mountainous stream in Korea, macroinvertebrate communities in the periods with the three-day cumulative rainfall as small as 100 mm or less and the dry periods did not vary significantly, whereas the decrease of the species richness and abundance was relatively much more when the rainfall exceeded 200 mm. The authors concluded that this observation was concurrent with IDH that an intermediate disturbance may increase the diversity of an animal community [58] since the three-day cumulative rainfall of 100 mm or less may belong to a normal disturbance category. Our findings, in a similar fashion, are consistent with IDH postulates in which biodiversity may increase due to an ecological disturbance of moderate magnitude.

The most flood-affected family was Gastropoda. This group was found to be virtually absent in the July 2020 sampling, followed by a steady recovery across all sites in the August survey. This sensitive nature of Gastropoda to hydrological variability is also consistent with earlier studies such as [59], who also reported a substantial decline in snail population following a flood event in Sycamore river, Arizona. On the other hand, a

higher abundance of Gastropoda during the low-flow periods is indicative of their ability to sustain lentic conditions that are often associated with very low flow, including elevated water temperature [60]. Taxa belonging to the Leptophlebiidae family (e.g., *Paraleptophlebia submarginata*) and Chrysomelidae family (e.g., *Donacia crassipes*) were not found in any of the post-flood samplings, possibly indicating a much longer time required for recovery.

Ecological community response to floods is highly context-dependent because of the non-linear and complex nature of ecological systems [61]. In the upper Jeziorka, biological diversity indices did not deteriorate following the flood event (on the contrary, they slightly improved). A similar observation was reported in a case study in the James River, Virginia, U.S., where the biodiversity of the BM community in the spring season was either unaffected or, in some cases, increased following a flooding event [62].

Previous research identified DO as one of the main water quality parameters affecting BM community structure [63,64]. In the Jeziorka River, the highest values of DO were detected in the March sampling, contributing to the BM assemblage during the normal flow conditions. *Cheumatopsyche lepida* and *Asellus aquaticus* correlated well with high levels of DO, implying that higher levels of oxygen can facilitate feeding and reproduction [65]. DO rates are usually dependent on both flow velocity as well as stream temperature. While faster velocity is conducive for more regeneration, warmer stream temperature can cause a decline in DO level. During high flow conditions, DO levels dropped, which could be a reason for the significant decline in the overall abundance and richness. Gastropoda density was found to be positively correlated with DO concentration [66], and in the agriculture-dominated Ciechanowska Upland in Poland, they can be considered as biological indicators of DO in water [67]. However, our findings regarding the increased abundance of *Ephemera vulgata* suggest its adaptive ability to survive with low DO in these peak flow conditions. As noted by [68], *Ephemera vulgata*'s oxygen dependence value is not temperature-dependent, which explains the reason for its survival in the warmer water environment.

Our findings regarding $NH_4$-N show that it was one of the contributing factors toward the BM community structure in the Jeziorka catchment. The response of BM taxa to $NH_4$-N levels varied according to their different tolerance levels. As our results showed, $NH_4$-N concentrations gradually increased following the flood event. It might have caused a gradual reduction in Ephemeroptera and Trichoptera taxa, while the abundance of Chironomidae, Lymnaeidae, and Physidae increased [69–71]. We hypothesize that increased ammonium-nitrogen concentrations in the Jeziorka catchment were mainly due to upstream point sources, whose contribution is highest at low flows. Our results, particularly the relationship between the BM community and ammonium-nitrogen, reveal that anthropogenic activities may have affected the BM habitats with more point and non-point source pollutions bringing more nutrients in streams.

Sediment is the most commonly identified pollutant correlated with BM community deterioration in freshwater streams [72]. However, the lack of continuous monitoring makes it very difficult to capture the peaks of suspended sediment concentration. As flood events have the potential to flush material, thereby driving the peaks of sediment concentrations, monitoring sediment concentrations using turbidity sensors facilitates accurate quantification of sediment impact on BM [73]. Our results with regard to TSS concentration did not indicate any strong relationship with the BM community structure. These findings are not surprising, as a weak quantitative relationship (non-significant) between fine sediments and BM for Oregon streams was reported by [70] and by [74], who noted water chemistry as more important than sediments for BM community structure. Only a slight increase in TSS after the flood event might have contributed to higher abundance for some Diptera and Oligochaeta, as they are positively impacted by increased loads of fine sediment. However, for the EPT taxa that are negatively correlated with higher sediment loads, no such clear trend was noticed [75].

One of the limitations of this study was the lack of availability of more frequent BM monitoring data. While discharge and water quality were measured 12 times during the study period, BM abundance data were collected only three times. It is plausible that

some of the cause-effect relationships discussed earlier could have been corroborated as well as more insights regarding the recovery times could have been obtained with more frequent sampling. However, due to the unpredictable nature of floods, this problem is also apparent in many other studies trying to quantify the effect of floods on river biota [76–78]. Furthermore, there is no commonly used and approved experimental design for investigating the ecological effects of floods (partly due to their stochastic nature), as shown in a systematic review on ecological responses to hydrological extremes in Europe [7].

Moreover, it is often challenging to distinguish the impacts of hydrological disturbances and seasonality impacts on ecological communities. In particular, in the northern hemisphere, it is common to observe the peak emergence of aquatic insects in late spring [79], which may often lead to increased metrics in this season as compared to other seasons. Since our sampling design did not explicitly permit to analyze the seasonality aspect, we compared historical BM abundance data on the Jeziorka River from June 2016 and October 2019, collected by the Chief Inspectorate for Environmental Protection (GIOŚ) [80]. The total abundance of BM was significantly higher in June than in October, and the difference could be attributed to a large extent to the difference in abundance of one of the families of aquatic insects, i.e., Simulidae. Thus, the probable seasonal pattern is unlikely to explain the changes in the BM community documented in this study, as it goes in the opposite direction to the one that we have reported here. Similarly, the Liwiec river, which is located very close to Jeziorka, [81] reported a slight increase in abundance and richness of the BM community in the summer season compared to spring. Several other relevant studies reported similar observations for the U.K. [82] and northern Spain [83].

## 5. Conclusions

A long-lasting and periodically heavy rainfall that occurred in central Poland in June 2020 caused a rapid water level rise by almost 2 m and a nearly 10-fold discharge increase (the highest peak flow since 2010) in the low-gradient, temperate Jeziorka catchment. Thanks to the fact that comprehensive environmental monitoring had been set up in advance, the occurrence of this seasonally unpredictable extreme event offered a unique opportunity to study: (1) its effect on the BM community inhabiting a 1300 m long reach at the middle course of the Jeziorka River and (2) the potential effect of changing hydraulic and water quality conditions on the BM community and its habitat. We have found empirical evidence that:

(1) Taxa abundance and richness, as well as the ecological status of the reach measured by the BMWP-PL index, declined substantially following the flood, while no significant changes in Shannon evenness and Simpson diversity indices were observed.

(2) Community composition was strongly affected, but taxon-specific responses to the flood event were diverse. The least resistant taxa belonged to Gastropoda and Coleoptera, whereas *Cheumatopsyche lepida* and *Chironomus* sp. exhibited the highest resistance.

(3) The BM community recovered after the flood relatively quickly, as evidenced by sampling carried out 7 weeks after the flood peak, which suggests that most taxa showed high resilience to flood.

(4) The studied water quality parameters exhibited diverse responses during the flood. The most apparent ones were a decline in DO, pH, and $NH_4$-N and an increase in $NO_3$-N.

(5) Stream velocities exceeding critical velocity during the flood led to channel bed instability and increased bedload transport, as well as to a permanent modification of bed morphology as documented by depth sampling during discharge measurements.

It is difficult to infer cause-effect relationships between observed water quality patterns and BM community responses. However, it seems plausible that low DO values during the flood contributed to a decreased occurrence and abundance of some taxa such as *Planorbrius corneus* and *Valvata piscinalis*, whereas increasing $NH_4$-N concentrations after

the flood might have prohibited recovery of Ephemeroptera and Trichoptera taxa. On the other hand, the evidence that high stream velocities and increased bedload transport directly affected BM fauna seems strong.

Since the Jeziorka is a representative example of medium-sized, low-gradient rivers lying in the temperate zone and evidence on the effects of floods on BM communities in such rivers has been scarce to date, the results of this study can serve as the first approximation of the resistance and resilience of BM communities in similar river systems. Thanks to a wealth of quantitative data on both disturbance and abiotic and biotic response, the results can also serve as an input to the meta-analysis of ecological responses to floods [7,84]. Such empirical data can also contribute to the development of integrated modeling frameworks aimed at simulating the complex climate-flow-hydraulics-habitat-biota chains [85], which is of particular importance given the likely increase in the occurrence of flood conditions in Central Europe under a changing climate [86].

**Supplementary Materials:** The following are available online at https://www.mdpi.com/2073-444 1/13/7/885/s1, Table S1: Jeziorka taxon list, Video S1: Jeziorka flood.

**Author Contributions:** S.C.: conceptualization, methodology, data curation, software, formal analysis, writing—original draft, writing—review and editing. P.O.: data collection, laboratory analysis. A.K.: writing—review and editing. I.K.: laboratory analysis, editing D.M.-Ś.: data analysis, writing—review and editing. M.P.: funding acquisition, conceptualization, supervision, methodology, writing—review and editing. All authors have read and agreed to the published version of the manuscript.

**Funding:** This work is supported by the National Science Center (NCN) Poland under the research project called RIFFLES ('The effect of RIver Flow variability and extremes on biota of temperate FLoodplain rivers under multiple pressurES', grant 2018/31/D/ST10/03817).

**Institutional Review Board Statement:** Not applicable.

**Informed Consent Statement:** Not applicable.

**Data Availability Statement:** The data supporting the findings of this study are available from the corresponding author Mikołaj Piniewski upon reasonable request.

**Acknowledgments:** This work is supported by the National Science Center (NCN) Poland under the research project called RIFFLES ('The effect of RIver Flow variability and extremes on biota of temperate FLoodplain rivers under multiple pressurES', grant 2018/31/D/ST10/03817, https://projekty.ncn.gov.pl/index.php?projekt_id=422476, accessed on: 23 March 2021) The authors are grateful to NCN for providing the funding support. The Institute of Meteorology and Water Management–National Research Institute is kindly acknowledged for providing the flow data. The authors are also thankful to Robert Michałowski and student Michał Dytkiewicz for providing invaluable help with fieldwork at the Jeziorka and monitoring data treatment. We appreciate Damiano Baldan for his help with statistical analysis. We express our gratitude to the three anonymous reviewers and guest editors whose suggestions improved the quality of the manuscript.

**Conflicts of Interest:** The authors declare that they have no conflict of interest.

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
