# Peer review of "Effect of a Summer Flood on Benthic Macroinvertebrates in a Medium-Sized, Temperate, Lowland River"

_water, doi:10.3390/w13070885_

Round 1

Reviewer 1 Report

Review Feb 2021:

Effect of a summer flood on benthic macroinvertebrates in a medium-sized, temperate, lowland river

Somsubhra Chattopadhyay, Paweł Oglęcki, Agata Keller, Ignacy Kardela, Dorota Mirosław-Świątek and Mikołaj Piniewskia.

Objective: To quantify the effect of a moderate flood event that occurred in June and July of 2020 on the BM community structure in the Jeziorka, a typical medium-sized lowland river in Poland.

Main feedback:

  • The limited sampling periode, that various between season, is a challenge in order to conclude. However, the lack of studies evaluating summer flood impacts on BM is welcome. The authors should include more discussion on the uncertainty regard to this topic.
  • The lack of a proper presentation of the habitat features and analysis of the variation and impact on difference between habitat quality in BM sampling sites should be included. Substrate is a key variable and should be proper discussed as well. 
  • The paper needs to be restructured considering clear sub objectives and given hypotheses throughout the paper (all chapters). 
  • Is there a spatial and temporal scale issue in the paper? Spatial: the hydromorphological processes in a river is larger than the selected sampling scale, and the temporal scale considering BM community repsons to floods are limited. How does this impact the results?
  • Why is 10 year flood a proper flood for evaluating the impacts on BM community? Is it a large enough flood? what is the different impacts from various frequency floods? REferences and evaluation needed.

Does the introduction provide sufficient background and include all relevant references?

  • Introduction is well structured with large scale perspectives and the importance and role of BM as part of the ecosystem and their role.
  • The authors present large hydrological events such as floods as predominantly negative impacts on the lotic environment. However, large hydrological events are also important natural drivers for the aquatic community on stream systems. The authors are encouraged to also include thoughts and references on the natural role of large hydrological events as part of the climate change discussion, and balance both negative and natural/positive effects.
  • It is recommended that the authors broaden the text regard the role and importance of BM beside “mosaic of river ecosystem’s structure and as indicators of water quality (line 55). In what way is BM important for the ecosystem structure and water quality? Why? Examples, references?
  • How is climate change expected to impact Poland? References? The authors are encouraged to also include projected climate change impacts in Poland considering precipitation, temperature and water flow considering the study is from Poland.
  • The introduction includes an overall objective (“To quantify the effect of a moderate flood event that occurred in June and July of 2020 on the BM community”), however the authors need to include clear sub objectives and belonging hypotheses. These sub objectives and hypotheses should be the red line throughout the paper, including discussion.

Is the research design appropriate?

  • The research design is

Are the methods adequately described?

  • The section “study area” and “Flood event” should be relocated to the “Materials and Methods” chapter.
  • The description of the study area could include a table with key parameters/Features of the given area/river/Catchment, etc. E.g. features of the catchment area, river width/Depth/dominant substrate/mean annual flow/selected frequency floods, etc. A table will give the reader a quick overview of the type of catchment, hydromorphological features, aquatic community, etc. This is important to understand the study area and findings, and transferability to other studies (past and future).
  • The description of the flood event lacks information on frequency flood. Was it a 10 year flood? 100 flood? The title of the paper refer to “extreme”. What is an extreme flood in this Polish river?
  • The paper, including key words, introtext etc, refer to “extreme floods” and climate change with large hydrological events. However, according to the given picture (Fig.1) it is questionable whether this is an extreme or not, and how large the frequency flood is. Please ensure that the context of the paper is balanced with the event selected for analysis, and ensure to include information about the size of the frequency flood relevant to “extreme floods” and large floods etc.
  • Please provide more information on the RHH method, including uncertainties.
  • The BM sampling was conducted during one year in (March – August) whereas the flood event was in late June(?). Is the selected sampling period proper in order to evaluate the impact of a moderate summer flood on the BM community? Would the inclusion of several seasons or more years with sampling data affect results? The authors should include rationale and uncertainties regarding sampling periode and no. of samples according to main objective with the paper.
  • The sampling period is distributed between different seasons (march – august). Invertebrate sampling are sensitive between seasons considering species composition and number. How is this evaluated? Affecting the results? Rationale should be included as part of the method paper.
  • The author notes that the sampling sites differed in substrate conditions considering habitat quality. On the same time the authors concludes that the sampling sites are homogenous. However, substrate quality is probably the most important factor considering BM community. How is this affecting the results? The authors are encouraged to refer to standardized methods for BM sampling and whether the selected sampling method are in line with this. Also, the authors are encouraged to include a statistical test on the difference between the sampling sites based on systematically mapped habitat features. It should be evaluated to use ordination as an analytical technique that account for the multidimensionality of the data. This also reduces the chance of false positives (Type I errors), and involves Principal Components Analysis (PCA) or Correspondence analysis (CA).
  • Flood frequency analysis (FFA) is conducted. As mentioned above this is important to consider, however, as FFA is conducted it is maybe natural that this is part of the objective with the paper?
  • The river bed stability is calculated. Would this also be a sub objective? Evaluate whether the river bed is stable after a 10 yr summer flood, and the potential effects on BM community? It is recommended that the authors also refer to the paper by Hauer & Pulg (2018) considering river types, bed stability, morphological processes and classification as part of the evaluation of the main objective with this paper.
  • The river bed stability is evaluated, however, what role does the river edges/margins have? Important also considering the BM community and the selected objective of this paper?
  • Should normality tests be conducted? It is noted that e.g. Kruskal–Wallis rank sum is used, however, it is not noted whether the data is normally distributed or not. Please include a paragreaph on how and why the statistical tests were used.

        Are the results clearly presented?

  • The Result chapter should be structured according to selected sub objectives given in the Introduction chapter. Now, the authors have not included sub objectives with given hypothesis. It is recommended that the authors re-structure the paper, included the Result chapter according to the selected sub objectives and hypotheses.
  • The focus and results related to water velocity could be further developed by the inclusion of the importance of water velocity, or shear stress, along the river bottom. Is mean velocity a good enough variable and relevant for BM community? Why is high velocity negative for the BM community? Several papers indicate that mosaic habitat areas, e.g. riffles, support large variation in BM community.

      Author Response

We would like to thank Reviewer 1 for the time devoted to our manuscript and many useful comments. Please find below our point-by-point responses.

  • The limited sampling periode, that various between season, is a challenge in order to conclude. However, the lack of studies evaluating summer flood impacts on BM is welcome. The authors should include more discussion on the uncertainty regard to this topic.

Response: We appreciate the reviewer for this insightful comment. In the discussion section we now describe some additional points regarding the uncertainty (lines 496-506).

  • The lack of a proper presentation of the habitat features and analysis of the variation and impact on difference between habitat quality in BM sampling sites should be included. Substrate is a key variable and should be proper discussed as well. 

Response: While we agree with this great suggestion by the reviewer, due to non-availability of data it was not possible for us to conduct any statistical analysis such as CCA or PCA. We would like to stress that the habitat conditions were in general mostly homogenous with some minor differences. We have modified the text accordingly and have replaced “homogenous” with mostly homogenous substrate. We hope that there is more clarity regarding this substrate now. 

  • The paper needs to be restructured considering clear sub objectives and given hypotheses throughout the paper (all chapters). 

Response: Thank you for this excellent recommendation, we have now included sub-objectives and restructured the paper accordingly.

  • Is there a spatial and temporal scale issue in the paper? Spatial: the hydromorphological processes in a river is larger than the selected sampling scale, and the temporal scale considering BM community response to floods are limited. How does this impact the results?

Response: We understand the reviewer’s concern about the scale issue. However, we selected the 8 monitoring locations in order to capture a range of heterogeneity in the ecological community.

  • Why is 10 year flood a proper flood for evaluating the impacts on BM community? Is it a large enough flood? what is the different impacts from various frequency floods? REferences and evaluation needed.

Response: There are different ways to characterize an extreme hydrological event and for floods, return period is one such method. Here, in this paper we decided to use the return period to assess the severity of the flood event. However, as discussed by Piniewski et al. (2017) there is no such clear relationship among return period and associated impacts in the ecological community. The term ‘proper flood’ seems inadequate for us in this case. In our opinion, the flood event that happened in June and July enabled us with a good opportunity to investigate the response on ecological community due to the high frequency monitoring already in place. As much as we would like to both accommodate the reviewer and expand the level of detail in this portion, our view is that to do so would introduce an unbalanced level of treatment of the many components of the study and add to the length of what is already a quite long paper.

Does the introduction provide sufficient background and include all relevant references?

  • Introduction is well structured with large scale perspectives and the importance and role of BM as part of the ecosystem and their role.

Response: We thank the reviewer about this feedback.

  • The authors present large hydrological events such as floods as predominantly negative impacts on the lotic environment. However, large hydrological events are also important natural drivers for the aquatic community on stream systems. The authors are encouraged to also include thoughts and references on the natural role of large hydrological events as part of the climate change discussion, and balance both negative and natural/positive effects.

Response: This is an excellent suggestion and we agree with this recommendation. We have now added in the manuscript some relevant points regarding the positive effects of large hydrological events on the BM community. The following specific lines were now included in the manuscript “On a contrasting note, floods are also important natural drivers for the aquatic community on stream systems. In a very recently concluded study on multi-decadal effects of floods on BM community structure in the Murray River, Le et al. (2021) reported increasing abundance in all functional feeding groups. The authors attributed this phenomenon to an influx of organic matter of all sizes, from particulate organic matter to coarse and large woody debris following floods, that provide a food resource and/or habitat. A similar conclusion was drawn in the previous work by Le et al. (2020), in which it was observed that floods caused an initial decline in BM abundance and richness which was succeeded by a sustained increase persisting for over 25 years before returning to preflood levels.”

  • It is recommended that the authors broaden the text regard the role and importance of BM beside “mosaic of river ecosystem’s structure and as indicators of water quality (line 55). In what way is BM important for the ecosystem structure and water quality? Why? Examples, references?

Response: Great comment and we have now added some text in the manuscript following the reviewer’s suggestion. The following sentence is now added to the paper ‘BM have many advantages as good water quality indicators because they are diverse, ubiquitous, can be easily collected and are sensitive to a range of environmental and chemical stressors (Musonge et al., 2020)”

  • How is climate change expected to impact Poland? References? The authors are encouraged to also include projected climate change impacts in Poland considering precipitation, temperature and water flow considering the study is from Poland.

Response: We appreciate the reviewer for highlighting the importance of climate change projections for Poland. We agree to this recommendation and have added some additional text. Following specific lines were added ‘ Average annual temperature and precipitation are both anticipated to increase for Poland as reported by Mezghani et al (2017) with temperature increases between 1 – 4 °C accompanied by an increment  of 6 – 16% in precipitation. Floods are projected to increase in future thereby creating a challenge for flood risk reduction, water management, and climate change adaptation for Poland (Piniewski et al., 2017).

  • The introduction includes an overall objective (“To quantify the effect of a moderate flood event that occurred in June and July of 2020 on the BM community”), however the authors need to include clear sub objectives and belonging hypotheses. These sub objectives and hypotheses should be the red line throughout the paper, including discussion.

 Response: We concur with the reviewer’s opinion about including sub-objectives in the paper. In addition to the overall objective, we have now added three sub-objectives, in order to carry the message with more clarity. These lines were now added to the paper ‘In order to achieve the main objective, the following sub-tasks identified in this study were a) to determine the return period of the flood event that took place in June and July of 2020 b) to analyze the river bed stability conditions impacted by the flood event and c) to quantify the impacts on water quality parameters and d) to evaluate the flood impacts on BM community structure using abundance, richness and diversity indices.”

Is the research design appropriate?

  • The research design is

Are the methods adequately described?

  • The section “study area” and “Flood event” should be relocated to the “Materials and Methods” chapter.

Response: Agreed and moved.

  • The description of the study area could include a table with key parameters/Features of the given area/river/Catchment, etc. E.g. features of the catchment area, river width/Depth/dominant substrate/mean annual flow/selected frequency floods, etc. A table will give the reader a quick overview of the type of catchment, hydromorphological features, aquatic community, etc. This is important to understand the study area and findings, and transferability to other studies (past and future).

Response: This is a great comment and we now include a table with the key abiotic features of the Jeziorka catchment. We believe this would be helpful for the reader to get a quick overview.

  • The description of the flood event lacks information on frequency flood. Was it a 10 year flood? 100 flood? The title of the paper refer to “extreme”. What is an extreme flood in this Polish river?

Response: We understand that the reviewer may be actually referring to the keyword ‘extreme flood” and not the title as we described this event as a moderate summer flood. As mentioned, if only summer discharges are considered, it is indeed a 10-year flood but if the whole-year time series is considered then it becomes a 5-year event. Nevertheless, we have now replaced the keyword as well and appreciate the reviewer to point out this inconsistency in our wordings.

  • The paper, including key words, introtext etc, refer to “extreme floods” and climate change with large hydrological events. However, according to the given picture (Fig.1) it is questionable whether this is an extreme or not, and how large the frequency flood is. Please ensure that the context of the paper is balanced with the event selected for analysis, and ensure to include information about the size of the frequency flood relevant to “extreme floods” and large floods etc.

Response: From this comment it seems as if we have not conducted the flood frequency analysis, which is not the case. In section 3.1 we estimated the return period of the peak discharge to equal 5 years (or 10 years if the analysis is restricted to summer), as already mentioned above. We do not feel that there is any imbalance between the magnitude or severity of the studied event and the context of our manuscript.

  • Please provide more information on the RHH method, including uncertainties.

Response: We guess that the reviewer has River Habitat Survey (RHS) method in mind here. However, we feel that adding more details about this method would make the already lengthy manuscript even longer. We have not used this method in this study, but only cited a reference that assessed the habitat quality of Jeziorka river using this method. Thus we decided not to further elaborate on this method as it was outside the scope and focus of the performed study.

  • The BM sampling was conducted during one year in (March – August) whereas the flood event was in late June(?). Is the selected sampling period proper in order to evaluate the impact of a moderate summer flood on the BM community? Would the inclusion of several seasons or more years with sampling data affect results? The authors should include rationale and uncertainties regarding sampling periode and no. of samples according to main objective with the paper.

Response: Thank you for this great comment. We agree with the reviewer that including several years of data would make the comparison easier and the impacts of flood could be better understood. We now discuss some of the uncertainties in the discussion section.

  • The sampling period is distributed between different seasons (march – august). Invertebrate sampling are sensitive between seasons considering species composition and number. How is this evaluated? Affecting the results? Rationale should be included as part of the method paper.

Response: Excellent observation, we agree phenology is a crucial factor in ecology of these macroinvertebrates which should be taken into consideration. However, we would like to stress that, typically, except some aquatic insects, this seasonality aspect is not so prominent in lowland rivers such as Jeziorka. As our study focused on the whole BM assemblage and not just aquatic insects, we feel that our results reflect the impacts of flood mainly and are relatively unaffected by seasonal changes. Additionally, we analyzed historical BM abundance data from the Jeziorka (June 2016 and October, 2019) which basically revealed that higher abundance was observed in June than in October (see more the response to the Editor). This further validates our hypothesis that in this study floods were the reason macoinvertebrate indices declined in July sampling. Further, in a study from a nearby lowland river, (Liwiec river in Poland), Królak and KoryciÅ„ska (2008), reported a slight increase in BM abundance and richness in summer which is in agreement with our hypothesis that decline in BM community structure in Jeziorka was due to the flood and not the season. Similar observations were also noted by Medupin (2020) in the UK as well as in the Pas River basin, northern Spain by Álvarez-Cabria et al. (2011). Finally, we also think that the evidence collected from the water level changes, hydromorphological variations and changes in water quality parameter overwhelmingly point towards the flood impacts on BM community structure than anything else.

  • The author notes that the sampling sites differed in substrate conditions considering habitat quality. On the same time the authors concludes that the sampling sites are homogenous. However, substrate quality is probably the most important factor considering BM community. How is this affecting the results? The authors are encouraged to refer to standardized methods for BM sampling and whether the selected sampling method are in line with this. Also, the authors are encouraged to include a statistical test on the difference between the sampling sites based on systematically mapped habitat features. It should be evaluated to use ordination as an analytical technique that account for the multidimensionality of the data. This also reduces the chance of false positives (Type I errors), and involves Principal Components Analysis (PCA) or Correspondence analysis (CA).

Response: We understand that simultaneous occurrence of “heterogeneity” and „homogenous” sites may have created confusion and regret the inconvenience caused. What we wanted to highlight was that though the sites were rather homogenous there were some minor differences in substrate type at some sites. Predominantly sand, silt and mud were present as substrate type. While we would like to perform additional statistical analysis to differentiate different sites as suggested by the reviewer, unfortunately we don’t have the necessary data to conduct these analyses.

As regards the sampling method, the one that we used in this study was similar to the one reported by Wyżga et al. (2011, 2013, 2014).

  • Flood frequency analysis (FFA) is conducted. As mentioned above this is important to consider, however, as FFA is conducted it is maybe natural that this is part of the objective with the paper?

Response: Please see our response earlier in the Introduction section regarding sub-objectives. We now explicitly state flood frequency analysis as part of objective of the paper.

  • The river bed stability is calculated. Would this also be a sub objective? Evaluate whether the river bed is stable after a 10 yr summer flood, and the potential effects on BM community? It is recommended that the authors also refer to the paper by Hauer & Pulg (2018) considering river types, bed stability, morphological processes and classification as part of the evaluation of the main objective with this paper.

Response: We thank the reviewer for this suggestion. We included bed stability as one of sub-objectives and have cited this reference in the main text.  

  • The river bed stability is evaluated, however, what role does the river edges/margins have? Important also considering the BM community and the selected objective of this paper?

Response: Thank you for this very valuable comment. We realize that river banks are important for the BM community, but the available measurement data collected under difficult measurement conditions due to the flood event, allowed us only a rough estimation of the river bed stability.

  • Should normality tests be conducted? It is noted that e.g. Kruskal–Wallis rank sum is used, however, it is not noted whether the data is normally distributed or not. Please include a paragraph on how and why the statistical tests were used.

Response: We thank the reviewer for this suggestion. We have now checked our data for normality distribution and found that the data is indeed normally distributed which allows application of Kruskal-Wallis and Tukey tests. Previous studies such as Calderon et al (2017) have employed these tests to investigate flood impact on BM community structure. However, in order to not make the manuscript excessively lengthy, we decided not to add further text along the line.

Are the results clearly presented?

  • The Result chapter should be structured according to selected sub objectives given in the Introduction chapter. Now, the authors have not included sub objectives with given hypothesis. It is recommended that the authors re-structure the paper, included the Result chapter according to the selected sub objectives and hypotheses.

Response: We have now restructured the paper according to the reviewer’s recommendation.

  • The focus and results related to water velocity could be further developed by the inclusion of the importance of water velocity, or shear stress, along the river bottom. Is mean velocity a good enough variable and relevant for BM community? Why is high velocity negative for the BM community? Several papers indicate that mosaic habitat areas, e.g. riffles, support large variation in BM community.

Response: It could be assumed that the reviewer is perhaps pointing towards the impacts of fast velocity caused by the flood event. Actually we considered critical velocity and not the mean velocity with regards to the bed stability analysis and possible mechanism of BM community dislodgement. Whenever the mean velocity exceeded the critical velocity, we inferred about the impacts on the BM community. We would like to reiterate that by no means we claim that the high velocity alone was the sole negative factor for the BM community. In the introduction section we have now included some additional references explaining some positive impacts on BM community from flood events.

References

Álvarez-Cabria M, Barquín J, Juanes JA. Macroinvertebrate community dynamics in a temperate European Atlantic river. Do they conform to general ecological theory? Hydrobiologia 2011; 658: 277-291.

Bis B, Mikulec A. Przewodnik do oceny stanu ekologicznego rzek Polski na podstawie makrobezkrÄ™gowców bentosowych [The outline of the ecological status assessment of rivers in Poland based on the benthic macroinvertebrates assemblages], 2013.

Calderon MR, Baldigo BP, Smith AJ, Endreny TA. Effects of extreme floods on macroinvertebrate assemblages in tributaries to the Mohawk River, New York, USA. River Research and Applications 2017; 33: 1060-1070.

Królak M, KoryciÅ„ska, M. Taxonomic Composition of Macroinvertebrates in the Liwiec River and its Tributaries (Central and Eastern Poland) on the Basis of Chosen Physical and Chemical Parameters of Water and Season. Polish Journal of Environmental Studies 2008; 17: 39-50.

Le CTU, Paul WL, Gawne B, Suter PJ. Quantitative Flow-Ecology Relationships Using Distributed Lag Nonlinear Models: Large Floods in the Murray River Could Have Delayed Effects on Aquatic Macroinvertebrates Lasting More Than Three Decades. Water Resources Research 2020; 56: e2019WR025896.

Le CTU, Paul WL, Gawne B, Suter PJ. Insight into the multi-decadal effects of floods on aquatic macroinvertebrate community structure in the Murray River using distributed lag nonlinear models and counterfactual analysis. Science of The Total Environment 2021; 757: 143988.

Medupin C. Spatial and temporal variation of benthic macroinvertebrate communities along an urban river in Greater Manchester, UK. Environmental Monitoring and Assessment 2020; 192: 84.

Mezghani A, Dobler A, Haugen JE, Benestad RE, Parding KM, Piniewski M, et al. CHASE-PL Climate Projection dataset over Poland – bias adjustment of EURO-CORDEX simulations. Earth Syst. Sci. Data 2017; 9: 905-925.

Musonge PSL, Boets P, Lock K, Goethals PLM. Drivers of Benthic Macroinvertebrate Assemblages in Equatorial Alpine Rivers of the Rwenzoris (Uganda). Water 2020; 12: 1668.

Needham TH. The biology of mayflies. E.W. Classey Ltd., London 1935, Reprint 1972.

Piniewski M, Prudhomme C, Acreman MC, Tylec L, Oglęcki P, Okruszko T. Responses of fish and invertebrates to floods and droughts in Europe. Ecohydrology 2017a; 10: e1793.

Piniewski M, Szcześniak M, Huang S, Kundzewicz ZW. Projections of runoff in the Vistula and the Odra river basins with the help of the SWAT model. Hydrology Research 2017b; 49: 303-317.

Wyżga B, Amirowicz A, Oglęcki P, Hajdukiewicz H, Radecki-Pawlik A, Zawiejska J, et al. Response of fish and benthic invertebrate communities to constrained channel conditions in a mountain river: Case study of the Biała, Polish Carpathians. Limnologica 2014; 46: 58-69.

Wyżga B, Oglęcki P, Hajdukiewicz H, Zawiejska J, Radecki-Pawlik A, Skalski T, et al. Interpretation of the invertebrate-based BMWP-PL index in a gravel-bed river: insight from the Polish Carpathians. Hydrobiologia 2013; 712: 71-88.

Wyżga B, Oglęcki P, Radecki-Pawlik A, Zawiejska J. Diversity of Macroinvertebrate Communities as a Reflection of Habitat Heterogeneity in a Mountain River Subjected to Variable Human Impacts. Stream Restoration in Dynamic Fluvial Systems, 2011, pp. 189-207.

Reviewer 2 Report

Review comments on “Title: Effect of a summer flood on benthic macroinvertebrates in a
medium-sized, temperate, lowland river”

This study examined the responses of macroinvertebrates and physico-chemical habitat to a relatively large flood in temperate lowland river in Poland. This study provide important information on the topic in this targeted study region where existing relevant knowledge is relatively less, and thus has a potential as a good contribution to the field. However, it contains some issues in its design to address the question and needs to be substantially revised to be considered as a publication in the journal Water.

Major issue:

This study focused on macroinvertebrate responses. A fundamental flaw in its design is the timing of sampling and the interpretation of data only in relation to the flood. As well known, aquatic insects emerge as adults at some times of year and thus affect naturally communities observed in rivers independent of any external human activities or natural events including floods such as the one in this study. It appears that the author attributed community structural changes on benthos in March and July and August to the flood in June and recovery from the flood. June in northern hemisphere is known as among months peak emergence of aquatic insects are normally observed. I wonder how it was possible to separate the effects of such phenological process from the possible recovery processes of benthos from the flood. I do not think that authors discussed about this issue at all. Because of this fundamental problem in the sampling design, it is logically impossible to frame the study as it is done in its current form. Authors discussed about the challenges involved in sampling at appropriate timing because of unpredictability of floods. However, I do not think that this does not justify the way the results are interpreted in the current manuscript.

Minor issues:

L.150: Where in the cross-section was flow measured?

L.195: One sample from each site? What are the size criteria of collected invertebrates? Are the size criteria consistent across the sites and occasions? Was sieving involved to fraction sizes? All these information descriptions should be more described.  

L.213: event

L.246-248: how and where in the channel was this collected? What is the replicate at each site?

L.255: ln(s)

L.265: was there any description that R was used?

L.273-291: To me, most of the information herein as well as figure 4 should be proved in the method because the objective of the study is not to characterize the flood but to examine the response of invertebrates.

L.288: It was unclear how the velocity measurement described in the methods can provide this figure

Fig.4: C is not necessary as it does not provide any additional information. Also, date/year expressions are different among figures and need to be unified for clarity. Throughout the manuscript, figure quality is not acceptable in terms of consistency. Numbers that should be described as superscript are, for example, not shown that way in many places.

Fig.8: Please describe site name where this measurement was taken.

L.341: PO4-P; this kind of inconsistency can be seen in places, for example, in the axis labels and figure titles in figure 7

Fig.7: caption should at least say which of bar or line plots denote flow

L.352- and methods about the statistics: from the results and descriptions of analytical approach (L.267-), it is very unclear how ANOVA was used in the analyses. Descriptions should be clear so that how main effects in ANOVA was tested and its results first followed by presentation of results of multiple comparisons.

L.431: It is unclear why the author argue that BM composition change was largely caused by velocity based on the findings presented…when multiple other environments

has also changed.

Author Response

Response: We would like to thank Reviewer 1 for the time devoted to our manuscript and many useful comments. Please find below our point-by-point responses.

This study examined the responses of macroinvertebrates and physico-chemical habitat to a relatively large flood in temperate lowland river in Poland. This study provide important information on the topic in this targeted study region where existing relevant knowledge is relatively less, and thus has a potential as a good contribution to the field. However, it contains some issues in its design to address the question and needs to be substantially revised to be considered as a publication in the journal Water.

Response: Thank you for your positive general feedback of our manuscript.

 Major issue:

This study focused on macroinvertebrate responses. A fundamental flaw in its design is the timing of sampling and the interpretation of data only in relation to the flood. As well known, aquatic insects emerge as adults at some times of year and thus affect naturally communities observed in rivers independent of any external human activities or natural events including floods such as the one in this study. It appears that the author attributed community structural changes on benthos in March and July and August to the flood in June and recovery from the flood. June in northern hemisphere is known as among months peak emergence of aquatic insects are normally observed. I wonder how it was possible to separate the effects of such phenological process from the possible recovery processes of benthos from the flood. I do not think that authors discussed about this issue at all. Because of this fundamental problem in the sampling design, it is logically impossible to frame the study as it is done in its current form. Authors discussed about the challenges involved in sampling at appropriate timing because of unpredictability of floods. However, I do not think that this does not justify the way the results are interpreted in the current manuscript.

Response: We appreciate the thought-provoking observation from the reviewer and agree that phenology is an important issue. While we acknowledge the fact that peak emergence of aquatic insects is quite significant in June in northern hemisphere, our focus was not only limited to the aquatic insects such as Mayflies. During our field studies we have not observed the mass swarming (emerging flies phase) of any species of Ephemeroptera or any other invertebrate order, thus it is unlikely that the mentioned phenomena could impact the results of the sampling in the period when the larvae were caught.  The “classic” literature sources – especially related to small rivers – point out that adults of mentioned taxa in small lowland rivers have generally a long period of emergence (they may fly out in spring, summer or fall; (Needham, 1935, Reprint 1972)). The abundance of mayflies (and other invertebrates associated with mass swarming, such as caddis flies) on the studied reach was relatively low. In our analysis we looked at the entire macroinvertebrate community structure including insects. From our experience with biological sampling at Jeziorka and nearby rivers, there is no such strong seasonality component regarding the macroinvertebrate emergence at any particular time of the year. Thus while we agree with reviewer that peak emergence could be noticed very easily at other places, unfortunately we are not sure this is the case here, given the evidence seems mixed. We now address some of the uncertainty related to the sampling in the text (Lines 496-506).

Minor issues:

L.150: Where in the cross-section was flow measured?

Response: Flow was measured at site M8, it is now included in the text.

L.195: One sample from each site? What are the size criteria of collected invertebrates? Are the size criteria consistent across the sites and occasions? Was sieving involved to fraction sizes? All these information descriptions should be more described.  

Response: Yes, one sample was collected from each site, with a sampling area of 1m2 which was consistent across all sites and occasions. No specific size criteria were applied. The sampling equipment, i.e. mosquito dipper and a triangular dip net had a mesh size of 60 µm. In the laboratory phase, collected organisms were not fractioned by size.

L.213: event

Response: Lines removed.

L.246-248: how and where in the channel was this collected? What is the replicate at each site?

Response: Samples were collected along a straight section of the river, from both sides of the river, and at several locations in the cross section, and were averaged by mixing.

L.255: ln(s)

Response: Done

L.265: was there any description that R was used?

Response: Yes, figure 9 is R output, reference to the package is now included.

L.273-291: To me, most of the information herein as well as figure 4 should be proved in the method because the objective of the study is not to characterize the flood but to examine the response of invertebrates.

Response: Section moved to methods section.

L.288: It was unclear how the velocity measurement described in the methods can provide this figure

Response: Figure 5 was obtained with Surfer software, we now included this in the text.

Fig.4: C is not necessary as it does not provide any additional information. Also, date/year expressions are different among figures and need to be unified for clarity. Throughout the manuscript, figure quality is not acceptable in terms of consistency. Numbers that should be described as superscript are, for example, not shown that way in many places.

Response: Thank you for this comment, we have now removed figure C. Regarding quality of figure, we suspect that the problem may be actually associated with this pdf generation process. We have made a note of this and perhaps this can be taken care during the galley-proof stage, if the manuscript is accepted for publication.

Fig.8: Please describe site name where this measurement was taken.

Response: Fig 8 compares the abundance, richness and diversity indices across all 8 sites.

L.341: PO4-P; this kind of inconsistency can be seen in places, for example, in the axis labels and figure titles in figure 7

Response: We have thoroughly checked the manuscript for all such occurrences and have now fixed it.

Fig.7: caption should at least say which of bar or line plots denote flow

Response: The caption says orange bars show the flow.

L.352- and methods about the statistics: from the results and descriptions of analytical approach (L.267-), it is very unclear how ANOVA was used in the analyses. Descriptions should be clear so that how main effects in ANOVA was tested and its results first followed by presentation of results of multiple comparisons.

Response: We performed a simple one-way ANOVA and , in order to eliminate confusion, we have now discarded ANOVA and now focus only on the two statistical tests. Please also see our response to Reviewer 1 about normality test.

L.431: It is unclear why the author argue that BM composition change was largely caused by velocity based on the findings presented…when multiple other environments has also changed.

Response: We are sorry for the confusion but we have not claimed that velocity changes in are solely driving the changes in BM community. Along with velocity, other parameters together were responsible for the changes in BM community structure. The reviewer is absolutely correct in bringing this up and the text here is now modified to convey the message clearly.

References

Álvarez-Cabria M, Barquín J, Juanes JA. Macroinvertebrate community dynamics in a temperate European Atlantic river. Do they conform to general ecological theory? Hydrobiologia 2011; 658: 277-291.

Bis B, Mikulec A. Przewodnik do oceny stanu ekologicznego rzek Polski na podstawie makrobezkrÄ™gowców bentosowych [The outline of the ecological status assessment of rivers in Poland based on the benthic macroinvertebrates assemblages], 2013.

Calderon MR, Baldigo BP, Smith AJ, Endreny TA. Effects of extreme floods on macroinvertebrate assemblages in tributaries to the Mohawk River, New York, USA. River Research and Applications 2017; 33: 1060-1070.

Królak M, KoryciÅ„ska, M. Taxonomic Composition of Macroinvertebrates in the Liwiec River and its Tributaries (Central and Eastern Poland) on the Basis of Chosen Physical and Chemical Parameters of Water and Season. Polish Journal of Environmental Studies 2008; 17: 39-50.

Le CTU, Paul WL, Gawne B, Suter PJ. Quantitative Flow-Ecology Relationships Using Distributed Lag Nonlinear Models: Large Floods in the Murray River Could Have Delayed Effects on Aquatic Macroinvertebrates Lasting More Than Three Decades. Water Resources Research 2020; 56: e2019WR025896.

Le CTU, Paul WL, Gawne B, Suter PJ. Insight into the multi-decadal effects of floods on aquatic macroinvertebrate community structure in the Murray River using distributed lag nonlinear models and counterfactual analysis. Science of The Total Environment 2021; 757: 143988.

Medupin C. Spatial and temporal variation of benthic macroinvertebrate communities along an urban river in Greater Manchester, UK. Environmental Monitoring and Assessment 2020; 192: 84.

Mezghani A, Dobler A, Haugen JE, Benestad RE, Parding KM, Piniewski M, et al. CHASE-PL Climate Projection dataset over Poland – bias adjustment of EURO-CORDEX simulations. Earth Syst. Sci. Data 2017; 9: 905-925.

Musonge PSL, Boets P, Lock K, Goethals PLM. Drivers of Benthic Macroinvertebrate Assemblages in Equatorial Alpine Rivers of the Rwenzoris (Uganda). Water 2020; 12: 1668.

Needham TH. The biology of mayflies. E.W. Classey Ltd., London 1935, Reprint 1972.

Piniewski M, Prudhomme C, Acreman MC, Tylec L, Oglęcki P, Okruszko T. Responses of fish and invertebrates to floods and droughts in Europe. Ecohydrology 2017a; 10: e1793.

Piniewski M, Szcześniak M, Huang S, Kundzewicz ZW. Projections of runoff in the Vistula and the Odra river basins with the help of the SWAT model. Hydrology Research 2017b; 49: 303-317.

Wyżga B, Amirowicz A, Oglęcki P, Hajdukiewicz H, Radecki-Pawlik A, Zawiejska J, et al. Response of fish and benthic invertebrate communities to constrained channel conditions in a mountain river: Case study of the Biała, Polish Carpathians. Limnologica 2014; 46: 58-69.

Wyżga B, Oglęcki P, Hajdukiewicz H, Zawiejska J, Radecki-Pawlik A, Skalski T, et al. Interpretation of the invertebrate-based BMWP-PL index in a gravel-bed river: insight from the Polish Carpathians. Hydrobiologia 2013; 712: 71-88.

Wyżga B, Oglęcki P, Radecki-Pawlik A, Zawiejska J. Diversity of Macroinvertebrate Communities as a Reflection of Habitat Heterogeneity in a Mountain River Subjected to Variable Human Impacts. Stream Restoration in Dynamic Fluvial Systems, 2011, pp. 189-207.

Reviewer 3 Report

Chattopadhyay et al have performed a detailed and interesting work on the effect of summer floods on macroinverts. While the flood affected the community it was relatively rapidly recovered. The authors collected a very interesting dataset that covers various dimensions of floods.

I have few major concerns that may need to be considered before considering the manuscript for submission.

The introduction lacks of clear and specific hypothesis based on previous studies. Which is the actual knowledge gap that is going to be field by their work? And what is what they expect based on what is known?

Overall the figures are of low quality, it may be that the authors will provide better ones if the manuscript is accepted.

Minor comments

L33-34: The conclusion is too broad for the work performed, I suggest the authors to provide a more specific conclusion that can be linked with this general one.

L75-88: The work performed by the researchers is valuable regardless of the location of the river, especially because it is not a descriptive but a “manipulative” study. I do not see any added value to their work because little research has been performed in Poland. From a European perspective, results should be presented from any EU catchment comparable with Poland, for example any river that flows through Germany and Poland.

Sections 1.1 and 1.2 seem to rather belong to the methods section.

L269-270: I wonder whether a linear mixed model might be more suitable given that the measurements were performed in the same site at different times, so that using site as random factor. Or if not, a repeated measurements anova.

Author Response

Response: We would like to thank Reviewer 3 for reviewing our manuscript. Please find below our point-by-point responses.

Chattopadhyay et al have performed a detailed and interesting work on the effect of summer floods on macroinverts. While the flood affected the community it was relatively rapidly recovered. The authors collected a very interesting dataset that covers various dimensions of floods.

Response: Thank you for your generally positive feedback of the manuscript and associated dataset.

I have few major concerns that may need to be considered before considering the manuscript for submission.

The introduction lacks of clear and specific hypothesis based on previous studies. Which is the actual knowledge gap that is going to be field by their work? And what is what they expect based on what is known?

Overall the figures are of low quality, it may be that the authors will provide better ones if the manuscript is accepted.

 Response: We thank the reviewer for his time and opinion. We have now restructured the manuscript including sub-objectives and results section based on reviewer 1’s recommendation. We believe that there is much more clarity in the overall message now.

Minor comments

L33-34: The conclusion is too broad for the work performed, I suggest the authors to provide a more specific conclusion that can be linked with this general one.

Response: We respectfully disagree with the reviewer. We believe that the evidence we gathered in support of a prominent impact caused by the floods on the BM community was very apparent. This results should indeed help to understand BM resistance and resilience in temperate, lowland rivers.

L75-88: The work performed by the researchers is valuable regardless of the location of the river, especially because it is not a descriptive but a “manipulative” study. I do not see any added value to their work because little research has been performed in Poland. From a European perspective, results should be presented from any EU catchment comparable with Poland, for example any river that flows through Germany and Poland.

Response: We wanted to bring the local context of ecological response to hydrological disturbance here. The lack of studies in this area in Poland is definitely a motivation for this study. The handful of mentioned studies from Poland cited here, serves to justify the purpose of the study in our opinion.

Sections 1.1 and 1.2 seem to rather belong to the methods section.

Response: Agreed, done.

L269-270: I wonder whether a linear mixed model might be more suitable given that the measurements were performed in the same site at different times, so that using site as random factor. Or if not, a repeated measurements anova.

Response: We have now disregarded ANOVA to avoid confusion and focus only on the two different statistical tests to evaluate the flood impacts. 

Round 2

Reviewer 1 Report

The authors have improved the manuscript sufficient for publication. It is expected that the results and work accomplished should be of value to the scientific arena. We hope to see more studies on the given topic in future. 

Author Response

Thank you for your positive feedback and your earlier comments which helped to improve our manuscript in the previous round.

Reviewer 2 Report

Thanks a lot for taking carefully all the comments and suggestions

Author Response

(The authors gave the same response as above.)

Reviewer 3 Report

Overall the manuscript has been improved. I just have some concerns about the figures, they have different formats and quality. I would like to encourage the authors to submitted improved and uniform versions of the figures.

Author Response

Thank you for your positive feedback. We will take care of appropriate format and quality of figures upon proofreading stage.